# Sequential perturbations to mouse corticogenesis following in utero maternal immune activation

Cesar P Canales[1†], Myka L Estes[1†], Karol Cichewicz[1†], Kartik Angara[2], John Paul Aboubechara[1], Scott Cameron[1], Kathryn Prendergast[1], Linda Su-Feher[1], Iva Zdilar[1], Ellie J Kreun[1], Emma C Connolly[1], Jin Myeong Seo[1], Jack B Goon[1], Kathleen Farrelly[1], Tyler W Stradleigh[1], Deborah van der List[1], Lori Haapanen[3], Judy Van de Water[3], Daniel Vogt[2], A Kimberley McAllister[1‡*], Alex S Nord[1‡*]

[1]Center for Neuroscience, UC Davis, Davis, United States; [2]Department of Pediatrics & Human Development, Michigan State University, East Lansing, United States; [3]Division of Rheumatology, Allergy and Clinical Immunology, UC Davis, Davis, United States

**\*For correspondence:**
kmcallister@ucdavis.edu (AKMA);
asnord@ucdavis.edu (ASN)

[†]These authors contributed equally to this work
[‡]These authors also contributed equally to this work

**Competing interests:** The authors declare that no competing interests exist.

**Abstract** In utero exposure to maternal immune activation (MIA) is an environmental risk factor for neurodevelopmental and neuropsychiatric disorders. Animal models provide an opportunity to identify mechanisms driving neuropathology associated with MIA. We performed time-course transcriptional profiling of mouse cortical development following induced MIA via poly(I:C) injection at E12.5. MIA-driven transcriptional changes were validated via protein analysis, and parallel perturbations to cortical neuroanatomy were identified via imaging. MIA-induced acute upregulation of genes associated with hypoxia, immune signaling, and angiogenesis, by 6 hr following exposure. This acute response was followed by changes in proliferation, neuronal and glial specification, and cortical lamination that emerged at E14.5 and peaked at E17.5. Decreased numbers of proliferative cells in germinal zones and alterations in neuronal and glial populations were identified in the MIA-exposed cortex. Overall, paired transcriptomic and neuroanatomical characterization revealed a sequence of perturbations to corticogenesis driven by mid-gestational MIA.

## Introduction

Epidemiological association between maternal infection and neurodevelopmental disorders (NDDs) has been found for autism spectrum disorder (ASD), schizophrenia (SZ), bipolar disorder (BPD), anxiety, and major depressive disorder (MDD) (*Brown, 2011*; *Estes and McAllister, 2015*; *Parboosing et al., 2013*; *Patterson, 2009*). Indeed, maternal immune activation (MIA) itself is sufficient to produce NDD-relevant outcomes in offspring in animal models (*Canetta et al., 2014*; *Machado et al., 2015*). Offspring from female mice exposed to the toll-like receptor 3 (TLR3) agonist and viral mimic polyinosinic:polycytidylic acid [poly(I:C)], and offspring in maternal immune activation models using the bacterial mimic lipopolysaccharide (LPS) (*Depino, 2015*; *O'Loughlin et al., 2017*; *Simões et al., 2018*), recapitulate neuropathologies and aberrant behaviors seen in offspring born to flu-infected dams (*Shi et al., 2003*; *Smith et al., 2007*). MIA in mice produces behavioral and cognitive abnormalities with relevance across multiple NDD diagnostic boundaries (*Estes and McAllister, 2016*; *Malkova et al., 2012*; *Richetto et al., 2014*).

MIA-induced outcomes in adult offspring have been well characterized; however, mechanisms of initiation and progression of NDD-related brain pathology remain poorly defined. Poly(I:C) increases

proinflammatory cytokines such as IL-1β, IL-6, CXCL1, and TNF-α in the maternal circulation (*Ballendine et al., 2015*; *Dahlgren et al., 2006*; *Hsiao and Patterson, 2011*; *Meyer et al., 2006b*; *Meyer et al., 2006a*; *Smith et al., 2007*). Downstream pathological changes in the brains of offspring have been reported across cell types, from neurons to microglia, and these changes have been linked to perturbations in proliferation, migration, and maturation in the developing brain (*Haida et al., 2019*; *Shin Yim et al., 2017*; *Soumiya et al., 2011*; *Zhang et al., 2019*; *Zhao et al., 2019*). Recent studies have identified acute transcriptomic changes in fetal brain in mice and rats at single time-points (3–4 hr) following MIA, with significant overlap between transcriptome changes in the cortices of MIA-exposed offspring and those altered in the brains of children with ASD (*Lombardo et al., 2018*). Although MIA could cause a dynamic sequence of transcriptional changes in the fetal brain following acute exposure, no integrated picture of MIA-associated transcriptomic pathology exists that maps changes spanning from the initial acute response to MIA through birth. Here, we combined time-course transcriptomics with neuroanatomy to map changes in prenatal mouse cortical development following MIA induced by poly(I:C) injection at E12.5.

## Results

We used paired transcriptomic and neuroanatomical analyses to map changes in the fetal brain following MIA in a poly(I:C) mouse model (*Figure 1*, *Figure 1—figure supplement 1*, *Figure 1—figure supplement 2* and *Figure 1—figure supplement 3*). Induction of MIA via poly(I:C) injection at E12.5 produces relevant phenotypes in mice (*Choi et al., 2016*; *Shin Yim et al., 2017*; *Smith et al., 2007*) and the specific implementation of this model was recently validated in our hands to produce sufficient levels of MIA to mimic viral infection and cause aberrant behavioral outcomes in offspring (*Estes et al., 2020*). The dosage of 30 mg/ml poly(I:C) elicited substantial elevations in maternal serum IL-6 (on average 2200 pg/ml) 4 hr following injection (*Figure 1—figure supplement 1*). Pregnant mice were injected with saline or poly(I:C) at E12.5, and dorsal telencephalon (E12.5 + 6 hr, E14.5, E17.5, and birth (P0)) was microdissected (*Figure 1a*). RNA-seq datasets were generated from male and female embryos from 28 independent litters across control and MIA groups, typically with one to three embryos represented per independent litter (*Supplementary file 20*, *Supplementary file 5*, File source data 1). For control and MIA, 7 *vs.* 7 samples at E12.5, 13 *vs.* 13 samples at E14.5, 6 *vs.* 6 samples at E17.5, and 10 *vs.* 12 samples at P0, were compared in a differential expression (DE) analysis. For transcriptomic analysis, we used a strategy of first defining DE genes at each time-point using the general linear model (GLM) approach implemented in edgeR, followed by mapping DE signatures to systems-level expression patterns via module assignment using weighted gene co-expression network analysis (WGCNA).

Principal component analysis (PCA) of MIA and control RNA-seq samples showed developmental age accounts for the majority of variance across samples, with some additional separation between MIA and control groups (*Figure 1b–c*). We performed stage-specific differential expression analysis for male and female samples and for both sexes with sex as a covariate. For each time-point, DE genes were defined using a stringent false discovery rate (FDR) < 0.05 threshold and a more inclusive $p < 0.05$ threshold (*Figure 1d,e*, *Supplementary file 1–5*). Sex-stratified analysis indicated generally concordant DE differences between male and female samples following MIA, although DE effect sizes appeared stronger in females at E14.5 and E17.5, leading to a larger number of DE genes identified independently in females compared to males (*Figure 1—figure supplement 2*, *Supplementary files 6–15*). Overall, sex-stratified differences in DE genes were generally subtle, as demonstrated by high DE correlation and shared DE gene sets between sexes, with consistent findings across sexes among key DE genes (*Figure 1—figure supplement 3*). Samples from both male and female offspring were used for overall DE analysis models, with sex included as a covariate.

For the DE gene set passing the stringent FDR < 0.05 threshold, there were varying numbers of DE genes across time-points (*Figure 1d*). The strongest transcriptional signature was observed at E17.5 (2621 up and 3058 down), suggesting dramatic impact of MIA on cortical development at this time-point. P0 represented the subtlest transcriptomic signature, with no genes passing the stringent FDR < 0.05 threshold, and with the $p < 0.05$ DE genes showing a strong upregulation bias. We tested for overlap between DE RNA-seq genes and high confidence autism-associated genes in the Simons Foundation Autism Research Initiative (SFARI) gene database (*Basu et al., 2009*). The 82 mouse orthologs of SFARI ASD genes that were expressed at measurable levels were significantly

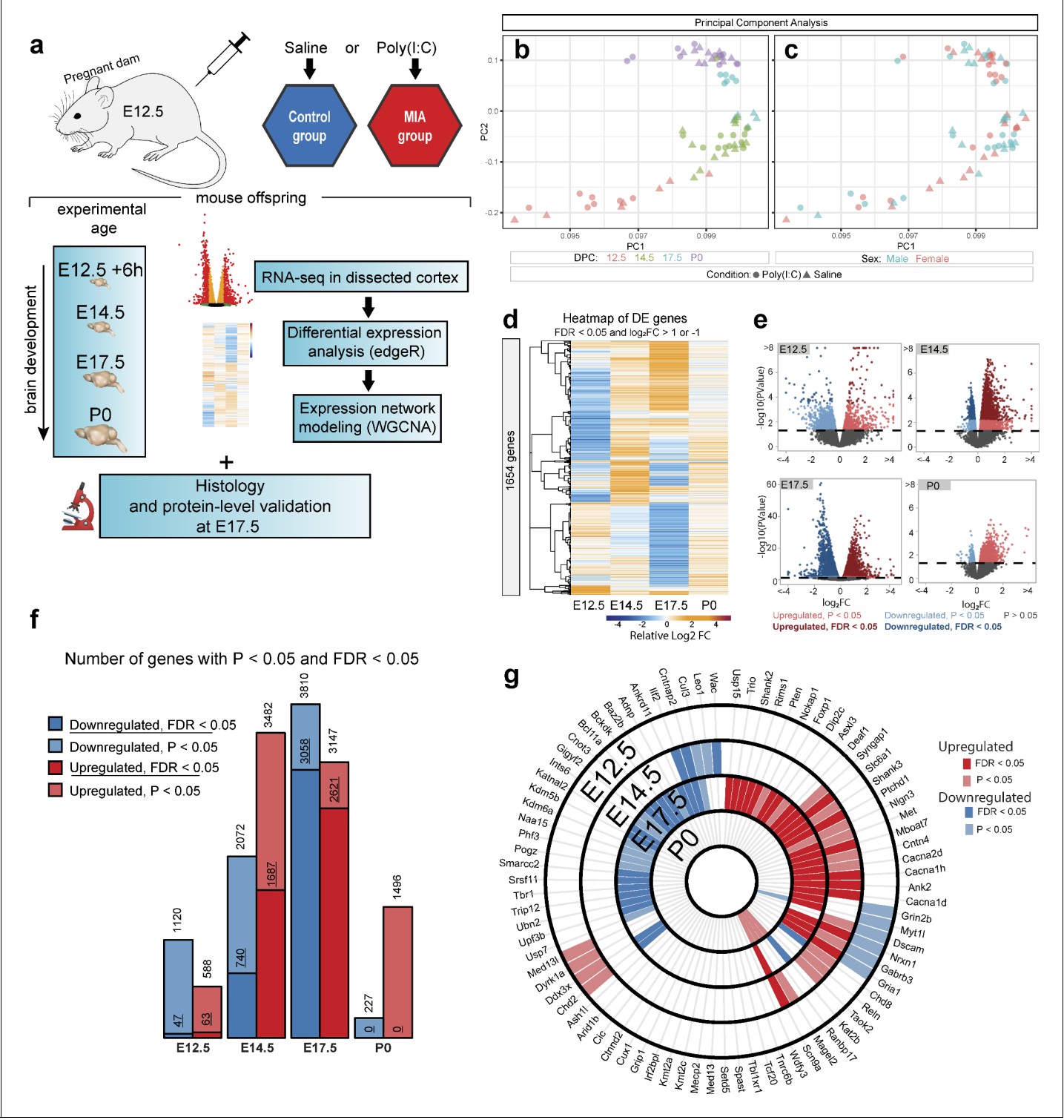

**Figure 1.** Differential expression across embryonic brain development following maternal immune activation via poly(I:C) injection at E12.5. (a) Schematic representation of MIA model and study design. E: embryonic day, P: postnatal day, IHC: immunohistochemistry, WB: western blot. Poly(I:C) was injected at E12.5. Samples for RNA-seq were collected at E12.5 + 6 hr, E14.5, E17.5, and at birth (P0). IHC and WB analysis were conducted on a separate animal cohort at E17.5 and P0. (b, c) Principal component analysis (PCA) of RNA-seq data indicates that developmental age accounts for the majority of variance across samples. Age (DPC) or sex are represented as colored symbols in (b) and (c) respectively, and poly(I:C) or saline treatment indicated by circles or triangles, respectively (d) Heatmap representing relative gene expression changes between control and MIA samples across time-points. Hierarchical clustering by relative fold changes shows stage-specific differential expression signatures (DE genes shown have FDR < 0.05

*Figure 1 continued on next page*

*Figure 1 continued*

and log$_2$ fold change (log$_2$FC)>1 or < −1). (**e**) Volcano plots of DE effect size and significance show stage-specific differences in number of DE genes and DE effect size, with the strongest dysregulation at E17.5. Colors represent directionality and statistical significance. (**f**) Numbers of upregulated and downregulated DE genes at FDR < 0.05 and p<0.05 thresholds again show varying DE genes numbers with a peak of dysregulated genes at E17.5. (**g**) Intersection of stage-specific DE genes with the 82 SFARI autism-associated mouse gene orthologs that were expressed at measurable levels in the RNA-seq data. E17.5 DE genes (FDR < 0.05) were enriched for SFARI genes (p=3.9e-04, hypergeometric test). Concentric circles represent developmental time-points. Light red, upregulated DE genes (p<0.05); dark red, upregulated DE genes (FDR < 0.05); light blue, downregulated DE genes (p<0.05); dark blue, downregulated DE genes (FDR < 0.05).

The online version of this article includes the following figure supplement(s) for figure 1:

**Figure supplement 1.** Validation of responsiveness to poly(I:C) based on dam serum IL-6 levels.

**Figure supplement 2.** Males and females show largely concordant DE signatures following MIA.

**Figure supplement 3.** Sex-stratified comparison of DE genes and effect sizes suggest increased DE effect sizes in females.

enriched among E17.5 DE genes, with 25 upregulated and 19 downregulated DE genes at FDR < 0.05 (p=3.9e-04, hypergeometric test), and 28 upregulated and 28 downregulated DE genes at p < 0.05 (p = 2.8e-06, hypergeometric test) (*Figure 1e*, *Supplementary files 16–17*). This enrichment during peak transcriptomic dysregulation suggests involvement of ASD-relevant pathways in MIA etiology here and demonstrates general NDD relevance of our model. Detailed results of DE analysis are reported in *Supplementary files 1–15* and can be visualized using our interactive online browser.

We next sought to capture MIA-induced gene regulatory changes at the systems and network level across embryonic cortex development using WGCNA (*Langfelder and Horvath, 2008*; *Zhang and Horvath, 2005*; *Figure 2*, *Figure 2—figure supplement 1*, *Figure 2—figure supplement 2*). In this approach, genes with correlated expression patterns are assigned into modules, enabling identification of gene sets with shared expression and function in neurodevelopment. WGCNA co-expression modules were arbitrarily named after colors, with Grey reserved for genes with no strongly correlated expression patterns (*Figure 2a*). Our initial analysis identified 10 modules (*Figure 2—figure supplement 1a*), of which correlated modules were combined to produce the final set of five modules and the unassigned Grey set. Modules were tested for association with sample age, MIA versus saline condition, and sex (*Figure 2b*). The modules captured dynamic trajectories of neurodevelopmental gene expression (*Figure 2c*). The Blue and Turquoise modules represented genes that increased or decreased in expression respectively, during neurodevelopment (Blue: p=9e-34; Turquoise: p=1e-29, *Figure 2—figure supplement 1b*). The other modules showed more complex patterns of expression. Each WGCNA module included DE genes, with differences in what time-point had the most extensive DE, indicating module-specific timing of perturbation associated with MIA (*Figure 2—figure supplement 1c*). DE genes were enriched for module-specific GO terms, with GO enrichment findings similar using all module genes and using an alternative rank-based gene set enrichment that is not dependent on FDR or p-value (*Mootha et al., 2003*; *Subramanian et al., 2005*; *Figure 2d*, *Supplementary file 18*; *Figure 2—figure supplement 2*, *Supplementary file 19*).

The WGCNA-resolved modules enabled mapping of DE signatures to neurodevelopmental processes that were acutely induced by MIA (*Figure 3*, *Figure 3—figure supplement 1*). There was significant association between genes in Grey (p = 0.007) and Green (p = 0.03) modules and MIA treatment, and marginal significance with MIA for the YeMaBl (p = 0.05) module (*Figure 2—figure supplement 1b*). The acute, initiating signaling pathways, found 6 hr after poly(I:C) injection captured by the Green module and by a subset of genes within the Grey module, show a strong signature of upregulated DE genes from E12.5 to E14.5 (*Figure 3a*). Analysis of the GO BP enrichment of upregulated DE genes in these modules identified activation of immune-related pathways, including defense response to virus, as well as angiogenesis and Vascular Endothelial Growth Factor A (VEGFA) signaling (*Figures 2c* and *3b*, *Figure 2—figure supplement 2a,c*). We tested for protein-protein association networks among the acute E12.5 DE (FDR < 0.05) genes using STRING (*Szklarczyk et al., 2019*), and found significantly more interactions than expected by chance (observed edges = 161, expected edges = 46, enrichment = 3.5, STRING p < 1.0e-16) (*Figure 3c*). This intersecting protein network was associated with metabolism, hypoxia, and stress (*Figure 3b*). The network includes a highly interacting core gene set with hub nodes, such as *Vegfa*, that have

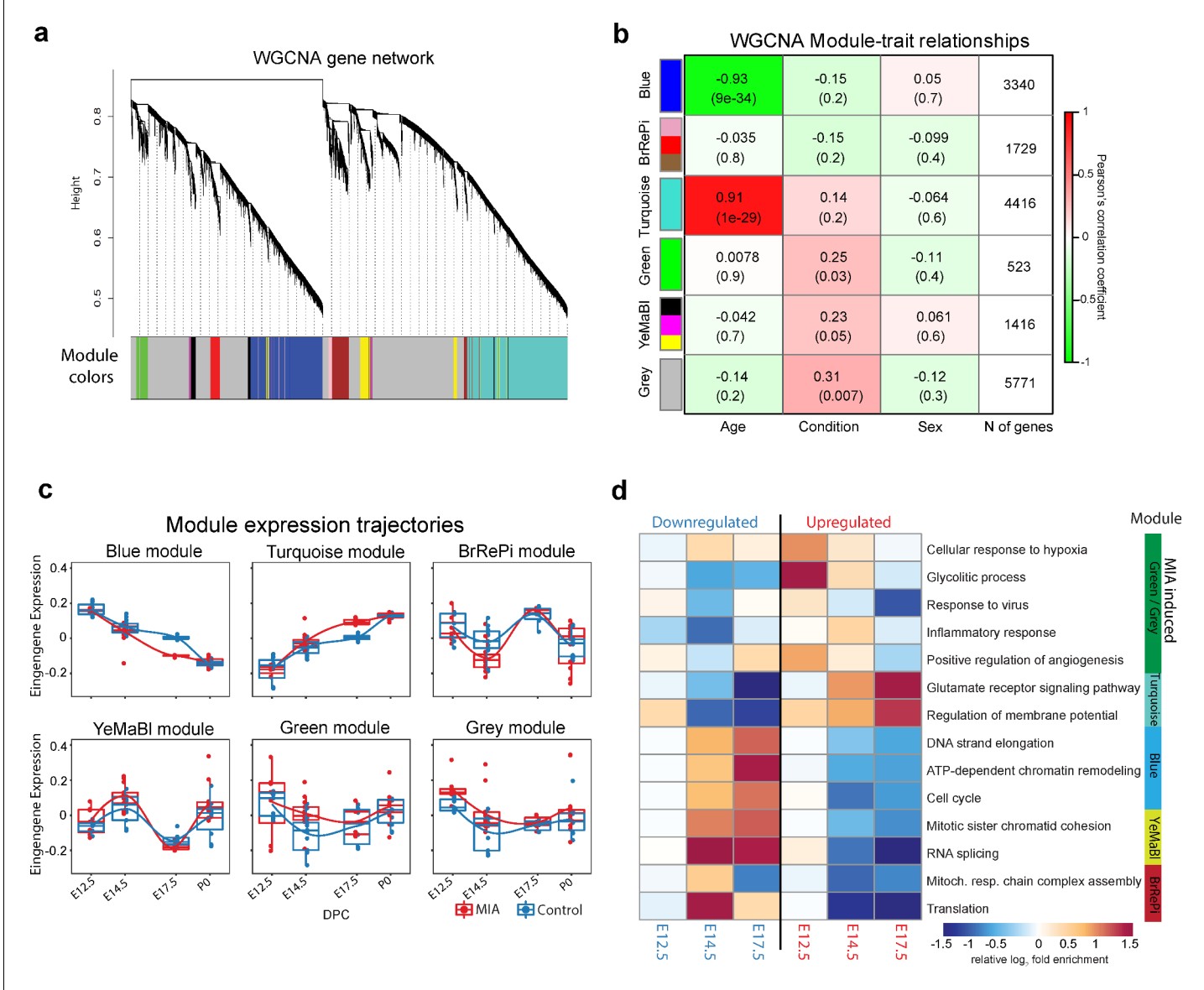

**Figure 2.** Weighted gene co-expression network analysis (WGCNA) reveals discrete gene networks perturbed by MIA across embryonic cortical neurodevelopment. (a) WGCNA cluster dendrogram of time series samples identifies genes that were assigned into co-expression modules. Based on similarities in module gene expression, six of the original 10 modules were grouped into two larger modules, BrRePi: Brown, Red and Pink; YeMaBl: Yellow, Magenta and Black. (b) Heatmap of correlation between gene expression modules and experimental traits; age, condition (saline vs poly(I:C)), and sex. Blue and Turquoise modules are strongly associated with age; Green and Grey modules are significantly associated with MIA condition. Numerical values represent signed Pearson's correlation coefficients, with Student asymptotic p values in brackets. Green represents negative and red represents positive correlation. Color intensity signifies the strength of the correlation. (c) Module eigengene expression for MIA and control groups plotted by time-point illustrates expression trajectories across developmental stages, capturing module- and stage-specific differences between MIA and control groups. (d) Heatmap of enrichment of DE genes for representative gene ontology biological processes (GO BP) by developmental time-point shows stage- and module-specific transcriptional pathology. Y-axis rows show enrichment for GO BP terms among module-specific DE genes (FDR < 0.05). Heatmap color scale represents relative fold enrichment for the GO terms among DE genes. P0 not shown due to insufficient DE gene numbers and absence of GO BP terms passing enrichment criteria.

The online version of this article includes the following figure supplement(s) for figure 2:

**Figure supplement 1.** Summary of WGCNA module eigengene expression, association with experimental variables, and overlap with DE genes.
**Figure supplement 2.** Stage-specific heatmaps of GO BP enrichment analysis of upregulated and downregulated differentially expressed genes in WGCNA modules.

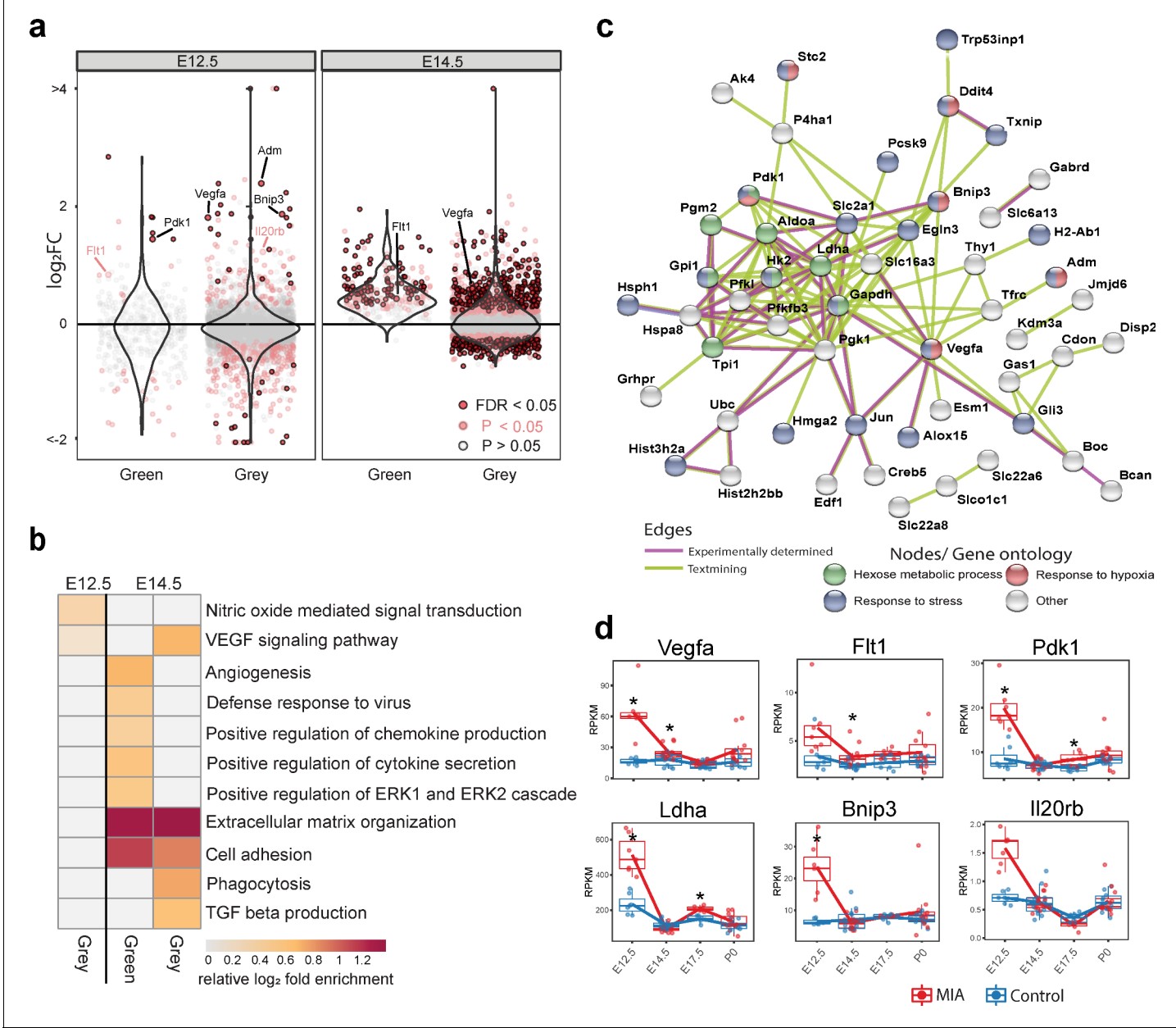

**Figure 3.** Green and Grey co-expression modules capture acute transcriptional response at 6 hr and 2 days following MIA exposure. (**a**) Violin plots visualize distribution of log₂ fold changes of Green and Grey module gene expression between control and MIA animals. At E12.5, an initial set of genes in the Green and Grey modules are induced and DE in MIA samples. By E14.5, generalized module expression exhibits induction in the MIA samples, with particularly strong change for the Green module, where nearly all genes are upregulated. Genes with expression trajectories shown in (**d**) are labeled. (**b**) Gene set enrichment analysis of GO BP terms significantly enriched among DE genes (FDR < 0.05) at E12.5 and E14.5 in Green and Grey modules showing upregulation of angiogenesis and immune pathways. Representative enriched GO BP terms with p < 0.05 colored by enrichment; gray represents enrichment p > 0.05 (Fisher's exact test). (**c**) STRING protein–protein interaction network of E12.5 DE genes (FDR < 0.05) colored by annotation to GO BP terms. There were significantly more interactions than expected by chance among these genes. (**d**) RPKM expression plots of genes that are associated with this network show acute DE at E12.5. Stars represent FDR < 0.05.

The online version of this article includes the following figure supplement(s) for figure 3:

**Figure supplement 1.** Sex-stratified RPKM gene trajectories of genes presented in the manuscript showing concordant effects between males and females.

many spokes and may be signaling factors that direct the MIA response (*Figure 3b,c*). Developmental expression profiles for DE genes that were associated with this interaction network, are shown in *Figure 3d*, and stratified by sex in *Figure 3—figure supplement 1a*. These signatures capture a complex but coherent transcriptional response involving hypoxia, immune, metabolic, and angiogenesis pathways that are strongly and transiently induced in fetal brain within 6 hr following mid-gestational MIA.

The most pervasive WGCNA-resolved DE signatures following MIA impacted a large proportion of genes in the Blue and Turquoise modules by E14.5 and peaked at E17.5 (*Figure 1c*). GO terms related to proliferation and cell cycle were enriched in the Blue module among downregulated DE genes, and terms related to neuronal differentiation and synapses were enriched in the Turquoise module among upregulated DE genes (*Figure 2c*). Considering the changes in proliferative and lamination markers associated with Blue and Turquoise genes, we sought to validate such changes at the protein level in independent samples (*Figure 4*, *Figure 4—figure supplement 1*). Four out of five DE genes tested validated with concordant changes at the protein level as determined by western blot (WB) analysis at E17.5, showing either statistical significance or trends in the same direction as observed in our transcriptomic analysis (*Figure 4a–b*, *Figure 4—figure supplement 1*). The DE signatures associated with Blue and Turquoise modules were evidenced by separation between MIA and saline samples in PCA plots at E14.5 and E17.5, with E17.5 MIA samples clustered closer to the P0 samples than saline age-matched counterparts (*Figure 1b*). To test for MIA-associated perturbation to temporal full transcriptome expression patterns, we generated a linear regression model using RNA-seq data principal components 1–5 and RNA-seq covariates to model age in saline samples. We then used this model to predict age in MIA samples (*Figure 4c*). E17.5 MIA samples exhibited older predicted age values relative to E17.5 saline controls (p = 4.5e-06, T-test), and E14.5 MIA samples showed a similar trend (p = 0.07, T-test). These findings suggest that mid-gestational MIA at E12.5 perturbs the timing of major downstream neurodevelopmental processes at E14.5 and E17.5 in embryonic cerebral cortex.

To characterize the neuroanatomical specificity of Blue and Turquoise dysregulation transcriptional signatures (*Figure 4d*), we performed a static immunofluorescent labeling comparison of late neurogenesis, cortical lamination and cell specification at E17.5 in an independent MIA cohort. E17.5 represents the end of the peak of neurogenesis during cortical development, before the neuronal-glial transition that occurs around E18.5. At this stage, it is expected that progenitor markers, such as PAX6, SOX9 and more specifically SOX2, will identify a small population of progenitors that are maintained and retain the ability to produce neurons, as well as subpopulations of proliferative cells commencing gliogenesis (*Martynoga et al., 2012*). PAX6+, SOX9+ and SOX2+ cell distribution patterns showed that progenitors localized correctly around the cortical ventricular zone in coronal brain sections; all analyzed cell populations were significantly reduced in MIA brains compared to controls (*Figure 4e–f*). The PAX6 finding in particular is consistent with our WB analysis at E17.5 (PAX6, p=0.039, two tailed Student's t-test) (*Figure 4b*, *Figure 4—figure supplement 1a*), with protein levels that either resolved or were too subtle to be distinguishable by birth (p=0.766, two tailed Student's t-test) (*Figure 4a*, *Figure 4—figure supplement 1b–c,i*). MIA brains showed a decrease in SOX9+ cells in the ventricular zone (VZ), which is expressed in progenitors (*Scott et al., 2010*). We also observed increased SOX9+ cells above the VZ in the neocortex, which can remain expressed in a subset of cells, including oligodendrocyte progenitors and maturing astrocytes (*Klum et al., 2018*; *Sun et al., 2017*) (SOX9 VZ, p=0.0198; nCtx, p=0.0058, two tailed Student's t-test). Reduced expression of SOX2, a progenitor marker that is restricted to the VZ, was also observed in MIA brains (SOX2 VZ, p=0.0006 two tailed Student's t-test) (*Figure 4f*). Overall, these data are consistent with the transcriptomic downregulation signature of proliferating cells in MIA brains.

To determine if the decrease in progenitor markers might coincide with decreased proliferation, we next labeled cells undergoing proliferation, using the marker KI67 (any stage of cell proliferation) and the Phospho Histone 3 (PH3) marker, which labels a subset of cycling cells in mitosis (M) phase (*Hendzel et al., 1997*; *Walton et al., 2019*). We also examined the post-mitotic intermediate progenitor marker TBR2. Analysis of KI67+ and PH3+ cells showed reduced numbers of labeled cells along with a qualitative reduction in numbers and ectopic positioning of postmitotic intermediate progenitor cells in the MIA group (TBR2+). Reduction in Ki67+ cells was observed in several regions of the developing cortex, including zones defined as S1 and S1DZ (*Choi et al., 2016*; *Shin Yim et al., 2017*), as well as the ganglionic eminences (*Figure 4e*). Quantification of Ki67+ cells and PH3

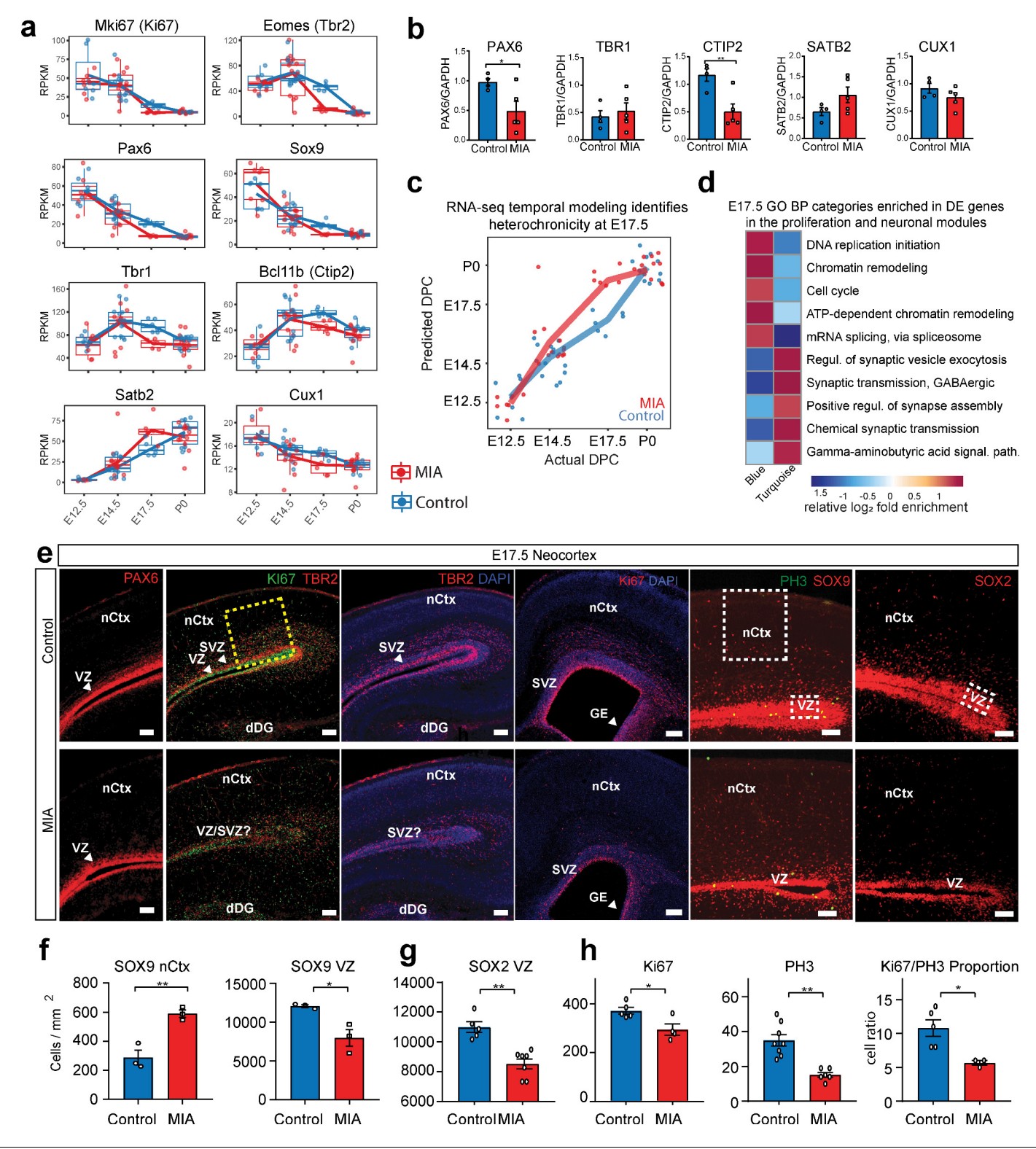

**Figure 4.** Altered cortical proliferation and lamination dynamics in fetal offspring five days after induction of MIA. (**a**) RPKM trajectories of proliferative and cortical lamination markers that were DE and validated at protein level. (**b**) Quantified protein levels from WB analysis validated DE changes at E17.5. Protein expression data are relative to GAPDH expression (n = 4 control, n = 5 MIA; PAX6 p=0.039, TBR1 p=0.615, CTIP2 p=0.0090, SATB2 p=0.1160, CUX1 p=0.2112; two tailed Student's t-test). Individual blots are in supplementary data (*Figure 4—figure supplement 1a–c,h–i*). (**c**) Temporal modeling of the RNA-seq data suggests acceleration of the neurodevelopmental program in MIA animals at E17.5 (p =4.5e-06, Student's

*Figure 4 continued on next page*

*Figure 4 continued*

t-test) and similar trend at E14.5 (p = 0.07, Student's t-test). Actual age (X-axis) vs predicted age (Y-axis) calculated by the linear model. Control samples were used for training the model. Lines connect average values, points depict individual samples and are jittered along the X-axis. (**d**) GO BP categories highlighting module-specific enrichment of E17.5 DE genes in the proliferation (Blue) and neuronal (Turquoise) WGCNA modules. (**e**) Representative images of coronal brain sections from E17.5 saline and MIA animals show reduced progenitor populations (neural stem cells: PAX6, SOX9, SOX2, intermediate progenitors: TBR2, and general proliferative markers: Ki67, PH3) and reduced active proliferation (Ki67:PH3 ratio). Staining for all cell populations is qualitatively decreased in the cortices from MIA offspring. Yellow box indicates the region of the neocortex utilized for Ki67 and Ph3 cell counts, and white boxes indicate regions of the VZ and upper layers of the nCtx utilized for SOX9 and SOX2 cell counts reported in (f-h). Scale bars = 100 μm, (labels: nCtx: neocortex, VZ: ventricular zone, SVZ: sub-ventricular zone, dDG: developing dentate gyrus, GE: ganglionic eminence). (**f**) SOX9 positive cell density in the VZ as well as in the upper layers of the nCtx shows reduced and ectopic SOX9+ cells in the MIA group (n = 3 per condition, SOX9 VZ *p=0.0198, SOX9 nCtx **p=0.0058, two tailed Student's t-test) (**g**) SOX2 positive cell density counts show a specific reduction in the VZ zone in the MIA group (n = 5 control group, n = 7 MIA group, SOX2 VZ **p=0.0006). (**h**) Ki67 and PH3 cell density quantification along the entire length of the VZ confirms this reduction in proliferative cells in MIA samples (Ki67 n = 5 control group, n = 4 MIA group, Ki67 *p=0.0198; PH3 n = 8 control group, n = 6 MIA group, PH3 **p=0.0003, two tailed Student's t-test) as well as a reduction in the Ki67/PH3 ratio (n = 5 control group, n = 3 MIA group, *p=0.0005 Chi-Squared).

The online version of this article includes the following figure supplement(s) for figure 4:

**Figure supplement 1.** Raw western blot and RNA-seq supporting data to proliferation, lamination, and cell-specificity analyses.

+ cells in the neocortex confirmed the MIA-induced reduction in progenitor and mitotic populations at E17.5 (Ki67, p=0.0198; PH3, p=0.0003, two tailed Student's t-test). The ratio of PH3+ (mitosis) cells to Ki67+ (all stages of cell division) cells was then used to determine whether cell cycle kinetics or the number of actively dividing cells in M phase was different in the MIA-exposed group. Ki67/PH3 ratios indicated a reduced proportion of progenitor cells in M phase in MIA brains at this specific time-point (p=0.0005 Chi-squared) (*Figure 4f*). These results provide evidence that the reduced proliferation signatures following MIA that were observed in the transcriptomic profiles are associated with changes at the protein level. Moreover, within the reduced progenitor cell populations at E17.5, the number of cells in the mitosis stage of the cell cycle are decreased to a greater extent.

MIA has been associated with altered cortical thickness and reported to cause cortical dysplasia in young adult MIA offspring (*Smith et al., 2007*; *Soumiya et al., 2011*; *Tsukada et al., 2015*), with stereotypic dysplasia impacting lamination localized to the somatosensory cortex reported in one MIA model (*Choi et al., 2016*; *Shin Yim et al., 2017*). A number of cortical layer markers were DE at E17.5 in RNA, with validated changes at the protein level (*Figure 4a–b*, *Figure 3—figure supplement 1b*). To examine cortical lamination, distribution of cortical layer-specific markers was analyzed via IHC in the same samples as above (*Figure 5*, *Figure 4—figure supplement figure 1 h-i*).

Examination of the distribution of the lamination markers TBR1 (layer 6), CTIP2 (layer five and a subset of cells in layer 6), SATB2 (layers 2–4, and a subset of cells in layer 5), and CUX1 (layers 2 and 3), revealed MIA-induced cortical layer-specific alterations in lamination, thickness, and cell density at E17.5 (*Figure 5a–b,e–f*). Overall, MIA brains appeared to be at a more mature stage of lamination at E17.5, showing a full separation of cortical deep layers more typical of later ages and in contrast to the partial layer overlap found in control offspring at E17.5 (*Figure 5a–b*). Similarly, SATB2 and TBR1 co-staining suggested differences in lamination and migration state between the two groups. Oval cell body shapes were observed, which resembled radially migrating cells (*Cooper, 2013*) were observed in both SATB2 and TBR1 cells in control tissue, whereas these cells showed a more rounded/post-migratory cell body shape in the MIA brains (*Figure 5c–d*). The number of TBR1+ and CTIP2+ cells, but not SATB2+ cells, were also significantly reduced in the neocortex from MIA offspring (TBR1+ cells, p=0.0138; CTIP2+ cells, p=0.0119; SATB2+ cells, p=0.306, two tailed Student t-test) (*Figure 5e*). The thickness of these layers and also a more external layer represented by CUX1, were measured by comparing the radial spread of stained cells within the neocortex. We found a significant reduction in CTIP2 layer thickness and a similar trend in TBR1 layer thickness in the MIA brains. In contrast, SATB2 showed an MIA-induced significant increase in layer thickness with no change in the more external layer represented by CUX1 (TBR1 thickness, p=0.081; CTIP2 thickness, 0.024; SATB2 thickness, p=0.039; CUX1, p=0.430, two tailed Student's t-test). (*Figure 5f*). These findings are consistent with individual trajectories of DE genes and WB analyses (*Figure 4a–b*, *Figure 4—figure supplement 1h*). Finally, to assess cortical gross anatomy, we looked for MIA-associated dysplasic alterations across the brain (*Choi et al., 2016*) and measured the thickness of the

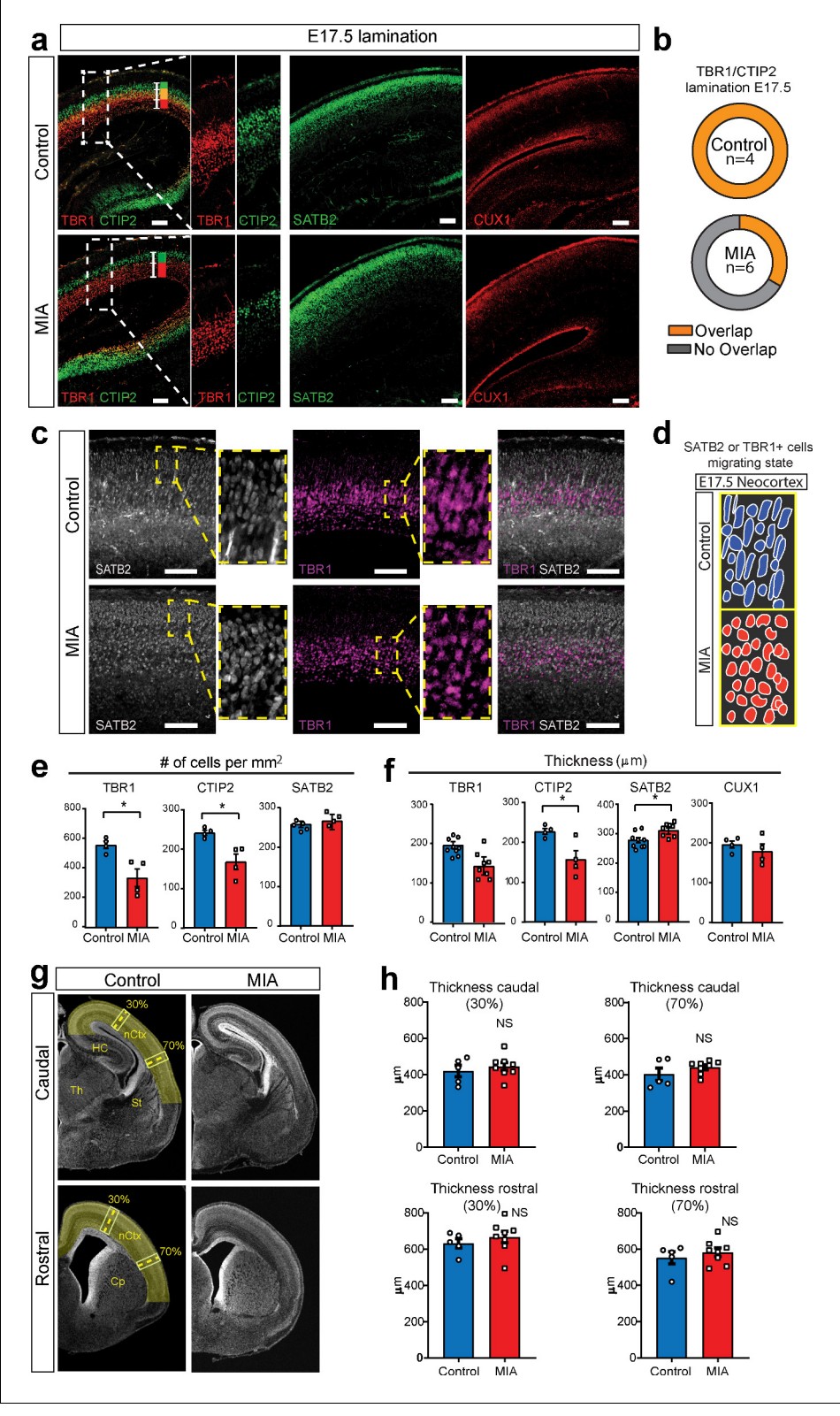

**Figure 5.** MIA impacts cortical lamination during development. (a) Coronal fetal brain sections from E17.5 saline (control) and MIA offspring show altered lamination pattern. Immunostaining using antibodies to identify deep cortical lamination layers using CTIP2 (green), and TBR1 (red) markers as well as markers of postmitotic neurons, SATB2 and CUX1, in more superficial cortical layers in brains derived from saline (control) or MIA injected dams.

*Figure 5 continued on next page*

*Figure 5 continued*

Representative images are shown. CTIP2/TBR1 overlap, represented by yellow color, was generally present in control brains, compared to fully laminated deep layers in MIA brains (no overlap illustrated by adjacent green and red rectangles). Boxes indicate magnified areas used for quantification and CTIP2/TBR1 overlap analyses. Scale bars = 100 μm. (b) TBR1/CTIP2 lamination overlap occurrence is present in all control brains but absent in the majority of MIA samples at E17.5. Brains from three MIA litters and two control litters were co-stained with TBR1 and CTIP2 cortical markers (as shown in a-b) and analyzed for lamination overlap occurrence (n = 4 control, n = 6 MIA). While all control brains showed lamination overlap between TBR1/CTIP2 layers, absence of TBR1/CTIP2 lamination overlap was observed in four out of six analyzed MIA brains. (c) Representative images of SATB2 and TBR1 staining at higher resolution show a change in cell morphology consistent with a difference in migration between MIA and control groups at E17.5. (d) Schematic representation illustrates the oval-radially migrating cell body shapes in control brains and the more rounded cell body shape in the MIA brains typical of static cells after migration. (e) MIA-induced reduction in TBR1+, CTIP2+, but not in SATB2+ cell counts (n = 4–5 brains per condition, TBR1 p=0.0138; CTIP2 p=0.0119; SATB2 p=0.306). (f) Significant decrease in CTIP2 and increase in SATB2 thickness indicates relative expansion of more superficial layers in MIA brains at E17.5 (Layer-specific thickness analysis: n = 4–8 per condition, TBR1 p=0.081; CTIP2 p=0.024; SATB2 p=0.039; CUX1 p=0.430, two tailed Student's t-test). (g) Representative caudal and rostral coronal sections of E17.5 control and poly(I:C) brains stained with DAPI show no obvious cortical dysplasia or gross anatomy alterations. Yellow shading represents neocortical areas considered for thickness measurements. nCtx: neocortex, Th: thalamus, St: striatum, HC: hippocampus, Cp: caudate-putamen. (h) Quantification of cortical thickness measured at 30% and 70% distance from the dorsal midline and cortical hemispheric circumference indicates no differences in overall thickness of these neocortical regions (n = 5 control, n = 8 poly(I:C); caudal 30% p=0.5252; caudal 70% p=0.2481; rostral 30% p=0.4459; rostral 70% p=0.4583, two tailed Student's t-test).

developing somatosensory cortex at two anteroposterior regions at E17.5. In contrast to previous reports (*Choi et al., 2016*), we did not find obvious cortical dysplasia or gross alterations in brain anatomy in the analyzed rostral and caudal regions, including somatosensory cortex and S1DZ (*Figure 5e–f*).

To investigate other cell populations that may be impacted by MIA during development, we analyzed astrocytes, oligodendrocytes and GABAergic interneuron cell populations (*Figure 6*, *Figure 4—figure supplement 1k*). Concordant with RNA-seq results, the number of cells expressing GFAP, a protein found in radial glia and a subset of astrocytes (*Eng et al., 2000*), was increased in MIA offspring, mapping specifically to cells that seemed to emerge ventral from the neocortical VZ (*Figure 6a–b,q*, *Figure 4—figure supplement 1k*). Since astrocytes also express SOX9 (*Klum et al., 2018*; *Sun et al., 2017*) and radial glia also express PAX6, which is decreased by MIA, this finding is consistent with the possibility that the increased SOX9+ cells in the dorsal regions of the neocortex, described earlier, may be indicative of early differentiated astrocytes (*Figure 4f*). OLIG2, which is expressed in oligodendrocyte progenitors and mature cells as well as transiently in astrocyte progenitors (*Dimou et al., 2008*), was also upregulated at E17.5 in the MIA group. OLIG2+ cells were increased in MIA brains compared to controls (n = 3 per group, p=0.0134, Student's t-test) (*Figure 6c–d,k*, *Figure 4—figure supplement 1k*). PDGFRα, a marker of early developing oligodendrocytes, was also increased in MIA brains compared to controls (n = 7 per group, zone 1 p<0.0001, zone 2 p=0.0063, zone 3 p=0.0125, Student's t-test) (*Figure 6e–f,l*). Thus, these experiments provide evidence that MIA leads to increased numbers of additional cell types (putative astrocytes and oligodendrocytes) that are typically born at later ages during corticogenesis, reinforcing findings of accelerated cell development that is supported by an increased representation of later-born cell types at E17.5 in brains exposed to MIA.

Finally, we examined DLX2 and GAD67, markers of committed and maturing cortical interneurons. Although *Dlx2* expression was downregulated in the MIA group from bulk tissue dissections at E17.5, the number of cells positive for DLX2 protein in the neocortex were comparable when assessed via IHC in the MIA and control groups (n = 3 per group, total DLX2 p=0.5494; by zones DLX2, zone 1 P=0.9848, zone 2 P=0.6903, zone 3 P=0.2714, Student's t-test) (*Figure 6g–h and m, o*). In contrast, GAD protein in this DLX2+ subpopulation, assessed by the pan-GABAergic marker GAD67+, was increased across the neocortex of MIA brains, and was not restricted to any specific zones of the neocortex (n = 3 per group, total DLX2+/GAD67 p=0.0004; by zones DLX2+/GAD67, zone 1 P=0.2277, zone 2 P=0.1352, zone 3 P=0.2862, Student's t-test) (*Figure 6i–j and n,p*). Since

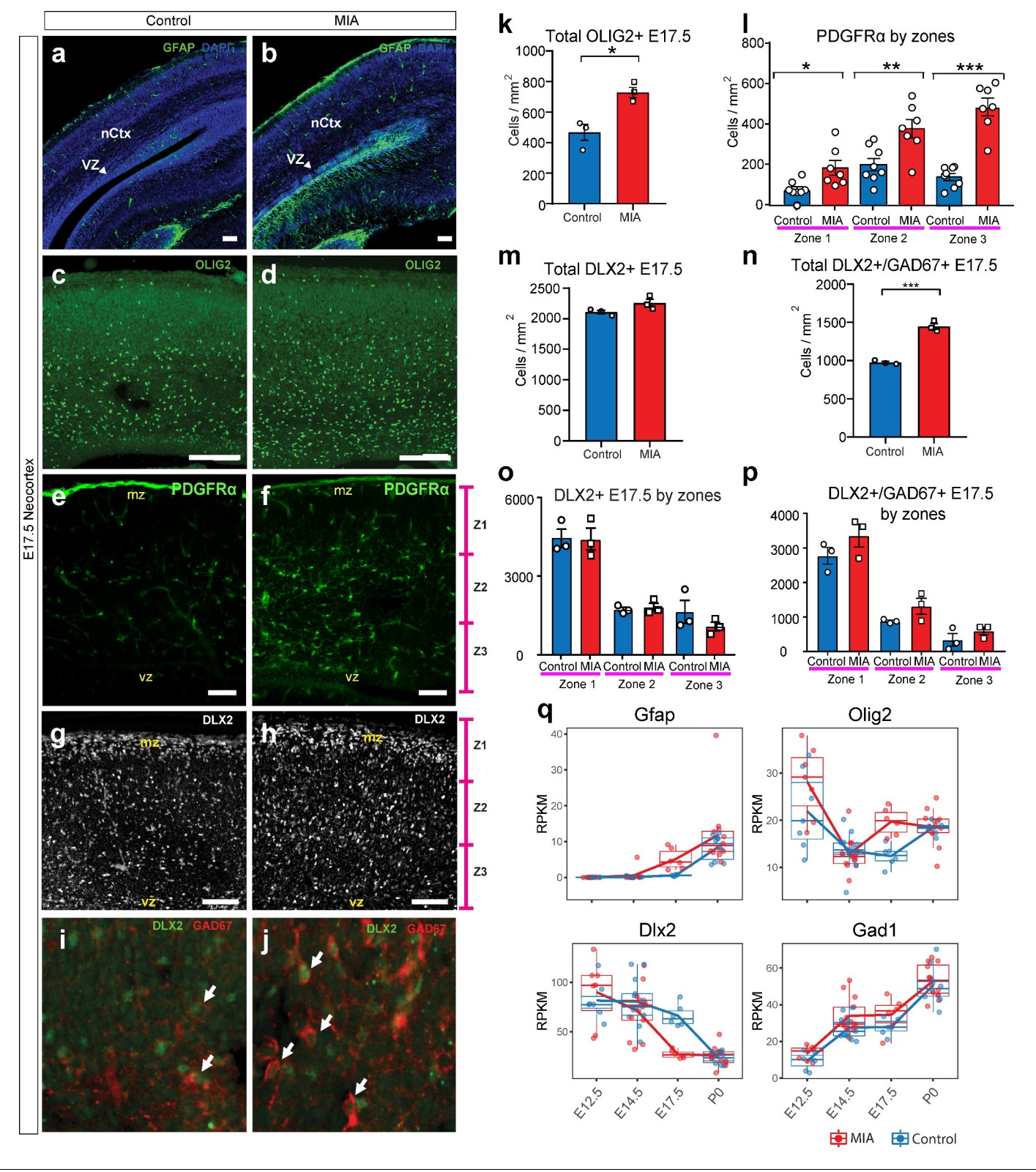

**Figure 6.** MIA alters development of neuronal and glial cell types in fetal offspring at E17.5. (**a–j**) Coronal fetal brain sections from three independent E17.5 saline litters (n = 4 control) and four independent MIA litters (n = 6 MIA) were immunostained for cell-type markers. (**a,b**) Altered distribution and numbers of GFAP+ cells (radial glia and astrocytes) in MIA samples. (**c-j**) Reduced sub-population of interneurons and oligodendrocytes using OLIG2, PDGFRα, DLX2, and GAD67. (**k-p**) Quantification of the cell density in (**c-j**). Overall MIA-associated increase in OLIG2+ cells (**k**), with further specific

*Figure 6 continued on next page*

*Figure 6 continued*

increased PDGFRα+ cells representing oligodendrocytes (**l**), and no change in overall DLX2+ interneurons (**m,o**), but increased overall number of DLX2 +/GAD67+ interneurons (**n,p**) are observed in MIA neocortex. Student's t-test: OLIG2 p=0.0134; DLX2+/GAD67+ cells p=0.0004. Bin quantification analyses of PDGFRα+, DLX2+, and GAD67+ immunoreactive cells in zones 1–3 (Z1 to Z3) indicate that changes are not restricted to a particular zones of the neocortex. Arrows denote co-labeled cells. Scale bars in (a-h) = 100 µm and in (i-j) = 30 µm. Abbreviations: neocortex (nCtx); marginal zone (mz); ventricular zone (VZ). (**q**) RPKM trajectories of the cell identity markers tested in (**a-k**) across the developmental time-points of the study show parallel differences in mRNA expression of cell-type markers following MIA induction. *Gfap* and *Olig2* mRNA was significantly upregulated in MIA samples at E17.5. *Dlx2* was significantly downregulated at E17.5, with no significant changes observed for *Gad1* (encodes GAD67) or for *Pdgfra* (not shown).

GAD67 expression increases in developing cortical interneurons, the decrease in *Dlx2* transcript expression but no difference in cells expressing DLX2 protein could indicate increased interneuron maturation but no change in interneuron numbers at this age. While neuroanatomical characterization here was limited to E17.5 and interrogated a subset of genes and processes of interest, our IHC findings correlate with transcriptional profiles, providing validation of MIA-associated DE in developing cortex.

## Discussion

Studies leveraging MIA models in mice have demonstrated the role of maternal immune signaling as a risk factor for cellular, anatomical, and behavioral outcomes in offspring. Nonetheless, a systems-level perspective of the early neurodevelopmental trajectory of pathology associated with MIA has been lacking. Our results offer a time course resolved map of molecular targets aberrantly transcribed in developing cerebral cortex following MIA. MIA-associated transcriptional signatures were robust, changed greatly across the surveyed time- points, and revealed novel pathways and processes involved in pathology. Among MIA-associated DE up-regulated and down-regulated genes, there was strong enrichment for ASD-relevant loci, indicating the general translational relevance of our MIA model to ASD. Both male and female samples were examined, and we did not find evidence for strong sex-specific DE signatures or impacts on neuroanatomy. Elucidating the trajectory of transcriptional changes across embryonic cortical development following mid-gestational MIA represents a valuable step toward understanding the progression of MIA-induced pathology in the developing brain of offspring.

Our results suggest two waves of transcriptional pathology following MIA. First, we found initial induction of immune, metabolic, and angiogenesis gene expression modules specific to MIA exposure and largely restricted to the days following MIA. There is overlap in the acute induction signatures identified here and acute changes reported in other mid-gestational MIA models, including those using a range of injection gestational ages and rat models of LPS (*Carpentier et al., 2013*; *Meyer, 2014*; *Meyer et al., 2006a*; *O'Loughlin et al., 2017*; *Richetto et al., 2014*; *Smith et al., 2007*; *Wu et al., 2017*). Previous studies linked maternal immune challenge, prenatal brain hypoxia, and impacts on proliferation (*Stolp et al., 2011*), neurogenesis, and cortical lamination (*Carpentier et al., 2013*). Thus, it is possible that the acute signatures identified here and mapped to the Green and YeMaBl modules as well as a subset of genes in the unassigned Grey module, capture a generalized response in fetal brain across mid-gestational MIA models. Notably, induction of hypoxia, cellular metabolism and neuroimmune transcriptional signatures, particularly captured here by the Green module, persist at reduced levels at least through P0, and may indicate lasting perturbation to these pathways. Significant association of the Grey unassigned module with the MIA phenotype was driven by a subset of genes that showed strong acute induction at E12.5 and E14.5. By altering WGCNA parameters, we could assign the Grey module genes to co-expression modules but those alterations increased splitting, resulting in a high number of modules and reduced module coherence without clarifying additional MIA-specific modules made up by the relevant Grey module MIA-associated genes. Thus, we opted for reduced splitting to capture major co-expression trends.

Identifying the early biological processes triggered by MIA are of critical interest for understanding how MIA initiates pathology. Our analysis identified potential drivers of these induced signaling pathways, including targets of substantial interest such as *Vegfa,* that link hypoxia, neurogenesis and cortical angiogenesis (*Argaw et al., 2012*; *Takahashi and Shibuya, 2005*). *Vegfa* was upregulated in MIA-exposed animal at all time-points and was among the strongest DE genes at E12.5 and P0.

*Vegfa* is critical for cortical development promoting neurogenesis and neurite outgrowth, as well as angiogenesis (*Bayless et al., 2016*; *Mackenzie and Ruhrberg, 2012*; *Rosenstein et al., 2010*). Upregulation of *Vegfa* is found in several NDDs and psychiatric disorders, including SZ and depression, and was also reported in a different MIA mouse model (*Arrode-Brusés and Brusés, 2012*; *Hohman et al., 2015*; *Iga et al., 2007*; *Isung et al., 2012*; *Kahl et al., 2009*; *Lee and Kim, 2012*; *Lizano et al., 2018*; *Xie et al., 2017*). Induction of angiogenesis in the developing cortex also independently causes a transition from proliferative state to neuronal maturation (*Bayless et al., 2016*; *Javaherian and Kriegstein, 2009*; *Tan et al., 2016*). Other genes of interest include *Pdk1* (pyruvate dehydrogenase kinase 1), which regulates glucose metabolism (*Newington et al., 2012*), *Ldha* (lactate dehydrogenase A), which facilitates glycolysis and participates in brain angiogenesis (*Lin et al., 2018*), *Bnip3* (BCL2 interacting protein 3), which is involved in cell death and mitochondrial autophagy (*Liu and Frazier, 2015*), and *Il20rb* (interleukin 20 receptor subunit beta), a receptor subunit for pro-inflammatory interleukins IL-19, IL-20 and IL-24 (*Akdis et al., 2016*). The early perturbed pathways observable within hours of MIA induction may be critical initiating steps leading to the altered general timing of fetal brain development described here. Our findings expand understanding of the molecular basis of these MIA-induced initial perturbations and reveal potentially key drivers of these changes.

The second component of the neurodevelopmental MIA signature defined here was perturbation to normal patterns of both proliferative and maturational neurodevelopmental modules that manifest several days after MIA induction. Initially, at six hours after MIA, there is a shift in expression of differentiation and neuronal maturation processes (e.g. synaptogenesis) to increased proliferative processes (e.g. positive regulation of cell cycle). This pattern parallels results described for the acute MIA response from other mouse and rat MIA models, where similar pathways were down- or up-regulated in fetal brain in the hours following MIA (*Money et al., 2018*; *Oskvig et al., 2012*). At this developmental stage, our data suggest that MIA induces an acute expenditure of the neocortical, and potentially other, progenitor pools, which in turn leads to an expedited maturation of cells that would normally be generated later during development. Specifically, by E17.5, multiple progenitor and cell cycle markers were significantly decreased in the MIA brains while concordantly, we observed an increase in maturational markers and laminar position that resembled a more developed cortex. Finally, astrocytes, oligodendrocytes, GABAergic interneurons and radially migrating excitatory neuron precursors were effected, suggesting an unbiased impact on progenitors of multiple brain cell types. Although these findings are indicative of overall decreased neurogenesis by E17.5, specific birth-dating and time-course experiments are needed to test whether progenitor cells exit the cell cycle differently and if overall neurogenesis is perturbed beyond the E17.5 time-point reported here. Nonetheless, our time course RNA-seq design revealed that the transcriptional changes following MIA can be separated into discrete biological and temporal expression modules that support altered proliferation and maturation of cell types following MIA.

Our findings of disrupted cortical lamination following the acute MIA signature are consistent with, but not identical to previous studies showing cortical dysplasia associated with MIA-induced aberrant behavior in offspring (*Choi et al., 2016*; *Shin Yim et al., 2017*). In those studies, dysplasia resolved at later postnatal stages but defects in lamination persisted into adulthood. Although we did not find evidence of aberrant cortical patches in the somatosensory cortex, secondary motor cortex or S1 and S1DZ, as previously reported, our results do show altered cortical lamination at E17.5. Our findings suggest that perturbation to neuronal migration, potentially due to differences in an early exit of pyramidal neuron precursors from the cell cycle that led to cells further along in their radial migration/lamination, developmental progression, at least in part contribute to MIA-associated lamination differences identified at E17.5 here. Thus, results from multiple studies using MIA models to date are consistent in showing a detrimental impact of MIA on cortical patterning; however, we also acknowledge that our and other studies may have differential variables, including onset of MIA, dose, mouse background, and endpoint measurements, which will be important factors to assess in future studies. Finally, bulk tissue RNA-seq is limited in cell type-specific resolution and signatures could additionally be in part driven by cortex-adjacent structures, such as primordial hippocampus, that may have also been captured at the youngest age. Future work using single-cell approaches is needed to resolve the impacted cell types and compare transcriptional pathology following MIA.

While significantly subtler in overall effects compared to earlier time-points, we also see evidence for altered neuronal transcription and lasting immune activation at P0. By P0, these signatures were reduced, which suggests a transient or attenuating effect on cortical patterning. It is also possible that biological variability in neonatal mice or technical variability impacted detection of gene expression changes in these P0 samples. Nonetheless, the subtle changes in gene expression at P0 are consistent with deficits in synapse formation that require changes in expression of immune molecules in neurons from newborn MIA offspring (*Elmer et al., 2013*).

Consistent with links between NDDs and developmental timing, atypical neurodevelopmental gene expression networks have been reported in a cortical neurodevelopmental model using autism-patient-derived induced pluripotent stem cells (*Bayless et al., 2016*; *Schafer et al., 2019*). The majority of the ASD-relevant genes that were DE following MIA (*Figure 1e*) were associated with perturbation to the modules associated with neurodevelopmental processes. These ASD-relevant genes map to functional and biological pathways generally associated with ASD and NDDs, including gene regulation, neurogenesis, and synapse development, structure, and function. The broad pathway overlap and presence of up- and down-regulation among DE ASD-relevant genes supports their generalized relevance. These overlapping genes represent targets of interest for future studies investigating parallel mechanisms between genetic and MIA-associated NDD risk. Our findings suggest some aspects of the transition from neurogenesis to astrogenesis appear to occur at an earlier stage following MIA based on similar signatures of SOX9+ and GFAP+ cells reported in literature (*Bansod et al., 2017*), and increased OLIG2+ cells and more mature GAD67+ DLX2+ interneurons in the cortex following MIA. These findings are additionally consistent with recent reports using poly(I:C) MIA models that identified perturbed GABAergic interneuron development (*Thion et al., 2019*), and maturation of GAD+ neuroblast after birth (*Vasistha et al., 2020*). Future work is needed to determine if and how MIA-associated dynamic transcriptional changes manifest postnatally and whether perturbation to embryonic development impacts cortical cytoarchitecture and function at later postnatal stages.

Altogether, our analyses indicate that MIA drives acute and lasting transcriptional changes that impact specific cell populations during forebrain development, disrupting normal developmental timing and causing increased numbers of astrocytes, oligodendrocytes, and altered molecular properties of GABAergic interneurons. The timing of MIA during gestation is known to cause a range of phenotypes in offspring, and thus the findings here regarding mid-gestational MIA at E12.5 may not generalize to other models with different timing of MIA induction, especially for later gestational MIA after neurogenesis is largely complete. Further, the strong embryonic signatures we observe were generally consistent across sexes, raising the question of whether secondary insults or sex-specific protective effects contribute to some reports of sex differences in pathology following MIA (*Haida et al., 2019*; *Naviaux et al., 2013*; *Smith et al., 2007*). Nonetheless, the overlapping results between our study and published single time-point studies of the acute MIA fetal brain transcriptional response suggest some consistency across bacterial and viral mimics, and rat and mouse models. Our study substantially expands previous work, revealing new molecular pathways and providing a temporal map of MIA-induced changes across fetal neurodevelopment that will be relevant to understanding the pathology of, and informing future treatments for, NDDs.

## Materials and methods

### Key resources table

| Reagent type (species) or resource | Designation | Source or reference | Identifiers | Additional information |
|---|---|---|---|---|
| Chemical compound, drug | poly(I:C) dsRNA | Sigma Aldrich | P0913 | Lot # 016M1451V |
| Antibody | Rabbit polyclonal anti PAX-6 | Covance | PRB-278P-100 | Dilution: (1:250) for IHC; (1:3000) for WB |
| Antibody | Rabbit polyclonal anti-TBR1 | Abcam | ab31940 | Dilution: (1:500) for IHC; (1:2000) WB |
| Antibody | Rat monoclonal anti-CTIP2 | Abcam | ab18465 | Dilution (1:250) for IHC; (1:1000) for WB |

*Continued on next page*

*Continued*

| Reagent type (species) or resource | Designation | Source or reference | Identifiers | Additional information |
|---|---|---|---|---|
| Antibody | Rabbit polyclonal anti-CUX1 | Abclonal | A2213 | Dilution (1:200) for IHC; (1:1000) for WB |
| Antibody | Mouse monoclonal anti-SATB2 | Abcam | Ab51502 | Dilution (1:500) for IHC; (1:2000) for WB |
| Antibody | Rabbit monoclonal anti-KI67 | Cell Signaling | 12202 | Dilution (1:500) for IHC |
| Antibody | Goat polyclonal anti-SOX9 | R and D Systems | AF3075 | Dilution (1:500) for IHC |
| Antibody | Rabbit polyclonal anti-PH3 | Cell Signaling | 9701 | Dilution (1:500) for IHC |
| Antibody | Rat monoclonal anti-TBR2 | Thermo Fisher Scientific | 14-4875-82 | Dilution (1:500) for IHC |
| Antibody | anti-DLX2 | John Rubenstein Lab | N/A | Dilution (1:200) for IHC |
| Antibody | Rabbit polyclonal anti-GFAP | Agilent Dako | Z0334 | Dilution (1:250) for IHC |
| Antibody | Rabbit polyclonal anti-OLIG2 | Millipore Sigma | AB9610 | Dilution (1:500) for IHC |

## Animal care and use

All studies were conducted in compliance with NIH guidelines and approved protocols from the University of California Davis Animal Care and Use Committee. C57BL/6N females (Charles River, Kingston, NY), were bred in house and maintained on a 12:12 hr light dark cycle at 20 ± 1℃, food and water available ad libitum. Mice utilized in this study do harbor Segmented Filamentous Bacteria (SFB) (*Estes et al., 2019*). Males and females embryos were analyzed across experiments, sex was determined as previously described (*McFarlane et al., 2013*).

## Maternal immune activation

Assessment of females baseline immunoreactivity before pregnancy and maternal immune activation were performed as previously described (*Estes et al., 2020*). Since female mice display a wide range of baseline immunoreactivity to poly(I:C) that dictates susceptibility and resilience of offspring from later pregnancies to MIA, virgin females in the lowest 25th percentile of immune responsiveness to poly(I:C) (IL-6 serum levels) were excluded from the study to reduce variability and ensure sufficient immune responsiveness of all included dams. E12.5 pregnancies were determined by visualizing vaginal seminal plug (noted as E0.5) and by body weight increase. IL-6 serum levels were measured at 2.5 hr post-injection. Temperature, weight, and sickness behavior were tracked as previously described (*Estes et al., 2020*).

## Tissue dissections for RNA-seq and immunoblotting

Dissections of dorsal telencephalon/pallium or developing cortex were performed at E12.5 + 6 hr, E14.5, E17.5, and P0 from control (saline) and poly(I:C)-exposed MIA litters. At E12.5, due to the smaller embryo size, dissections were made via a diagonal cut that maximized collection of dorsal telencephalon, but could have also included some subpallium/ventral telencephalon and superficial tissues. For later time-points, dissection was of pallium/developing cortex. Potential variance in the individual dissections should be mitigated by the inclusion of multiple replicates across both groups. Dissections included both hemispheres from males and females, with exact numbers and sample details reported in *Supplementary file 20*. Independent litters were used for RNA-seq and immunoblotting.

## RNA-seq

Following tissue collection as described above, total RNA was isolated using Ambion RNAqueous total RNA Isolation Kit and assayed using Agilent RNA 6000 Nano Bioanalyzer kit/instrument. Stranded mRNA libraries were prepared using TruSeq Stranded mRNA kits. Eight to 12 samples per lane were pooled and sequenced on an Illumina HiSeq 4000 instrument using a single-end 50 bp protocol. Reads were aligned to mouse genome (mm9) using STAR (version 2.4.2a) (*Dobin et al., 2013*) and gene counts produced using featureCounts (*Liao et al., 2014*) and mm9 knownGenes. Quality assayed using FastQC (*Andrews, 2010*) and RSeQC (*Wang et al., 2012*), and samples that

exhibited 3′ bias or poor exon distribution were discarded. Raw RNA-seq fastq files and a gene count matrix is available on GEO GSE166376.

## Immunoblotting

Tissue dissections from male and female littermates from at least three litters per condition were performed as described above and collected in HBSS, immediately frozen on dry ice, and stored at −80°C until processed. Samples were lysed in 50 mM Tris HCl, pH 8, 140 nM NaCl, 1 mM EDTA, 10% glycerol, 0.5% NP40 and 0.25% Triton with protease inhibitor cocktail (Roche). After sonication, samples were spun down and the supernatant was used for a BCA Bradford assay using the Spectra-max 190 plate reader to assess protein concentration using a standard curve. Twelve or 18 µg of protein were run on a 10 or 12% Tris acetate gel using the Mini-PROTEAN system (BioRad). Membranes were blocked in Odyssey blocking buffer (TBS; LiCor) and probed with the indicated primary antibodies overnight at 4°C. Resolved proteins were visualized in two channels using fluorescent secondary antibodies at 680 and 800 nm on an Odyssey Clx infrared imaging system (LiCor). Specific band intensities for all detected antibodies were quantified using the manufacturer's software ImageSoft (LiCor) and normalized to GAPDH loading control. Antibodies used were anti-PAX6 (1:250; cat #PRB-278P-100; Covance, Princeton NJ.), anti-TBR1 (1:500; cat#ab31940; Abcam, Cambridge, MA), anti-CTIP2 (1:250; cat# ab18465; Abcam, Cambridge, MA), anti-CUX1 (1:200; cat#A2213; Abclonal, Woburn, MA) and anti-SATB2 (1:500; cat#ab51502; Abcam, Cambridge, MA).

## Gross anatomy, immunohistochemistry, and image quantification analyses

Histology was performed in triplicate sections in brains using male and female pups from at least two litters. Sections were selected to focus on the developing somatosensory cortex. Details on sample individual sample usage for each IHC experiment is detailed in *Supplementary files 21* and *22*. Embryos used in IHC experiments came from independent litters versus RNA-seq and immunoblotting. Morphological parameters were measured using FIJI ImageJ (NIH), by an experimenter blinded to group and sex. Brain regions were identified based on anatomical landmarks as previously described (*Gompers et al., 2017*). No data points were excluded. For IHC staining, fetal brains were dissected and fixed with 4% paraformaldehyde/PBS solution overnight at 4°C. Tissues were then equilibrated in approximately 15 mL of 30% sucrose/PBS solution until they sank to the bottom of a conical tube. Equilibrated brains were embedded in Optimum Cutting Temperature (OCT) compound (Tissue-Tek, Torrance, CA) and frozen on dry-ice. OCT-embedded brain blocks were cryo-sectioned on a coronal plane (30 µm). Immunostaining was performed in free-floating sections with agitation. Sections were washed five times in PBST (PBS with 0.05% Triton X-100, 5 min each) and antigen retrieval was performed using 1x Citrate buffer pH6.0 antigen retriever solution (cat# C9999; Millipore-Sigma, Burlington, MA), at 60°C for 1 hr. Sections were washed five times in PBST (5 min each), permeabilized in PBS containing 0.5% Triton for 20 min and blocked for 1 hr at room temperature in 5% milk/PBST. Primary antibodies were incubated overnight at 4°C with orbital agitation (40–50 rpm). All antibodies used for this study were validated and their specificity validated. The following primary antibodies were used: anti-PAX6 (1:250; cat #PRB-278P-100; Covance, Princeton NJ.), anti-KI67 (1:500; cat#12202; Cell Signaling, Danvers, MA), anti-SOX9 (1:500 cat# AF3075, R and D systems, Minneapolis, MN), anti-PH3 (1:500 cat# 9701, Cell Signaling, Danvers, MA), anti-TBR2 (1:500; cat#14-4875-82; Thermo Fisher Scientific, Waltham, MA.), anti-TBR1 (1:500; cat#ab31940; Abcam, Cambridge, MA), anti-CTIP2 (1:250; cat# ab18465; Abcam, Cambridge, MA), anti-CUX1 (1:200; cat#A2213; Abclonal, Woburn, MA.), anti-SATB2 (1:500; cat#ab51502; Abcam, Cambridge, MA.), anti-DLX2 (1:200; generous gift from John Rubenstein, UCSF), anti-GFAP (1:250, cat#Z0334; Agilent Dako, Santa Clara, CA), anti-OLIG2 (1:500, cat#AB9610, MilliporeSigma, Burlington, MA). After primary antibody incubation, free-floating sections were washed five times in PBST (5 min each). Species-specific fluorophores-conjugated IgG (1:1000; Invitrogen-Thermo Fisher Scientific, Waltham, MA) were used as secondary antibodies (45 min, RT). 40,6-Diamidino-2-phenylindole (DAPI) (1:10000; Millipore-Sigma, Burlington, MA) was used for nuclear staining (20 min, RT). For cell counting, boxes were drawn on specific areas of the neocortex for progenitor and mitosis markers; SOX2, SOX9, and PH3. This box was 300 × 300 pixels on an image of the neocortex obtained using a ×20 objective. To be consistent, the ventral aspect of the box was drawn along the VZ and one

corner touched the medial aspect of the VZ. The same strategy was used to draw boxes from Ki67 images except the box was increased in size to 500 × 500 pixels to account for the additional Ki67+ cells seen in neocortical areas above the SVZ. SOX9 boxes were drawn along the dorsal border edge (600 pixels in width) of the neocortex and extended ventrally two thirds the length of the neo-cortex in order to avoid the VZ/SVZ region. For laminar zone counts, a box spanning 300 pixels was drawn along the VZ and the dorsal/ventral proportions were equally divided into three zones. These box sizes allowed for a sufficient number of cells to be counted from each tissue. Laminar thickness analyses were performed by measuring the spread of all positive cells within the cortical plate per individual marker. For all quantification analyses, one to three sections per brain utilizing the develop-oping hippocampus as a landmark for positioning were used.

## Bioinformatics analysis

Bioinformatic analysis was performed using R programming language version 3.5.1 (*R Development Core Team, 2015*) run in RStudio integrated development environment version 1.2.1269 (*Team R, 2018*). Plots were generated using ggplot2 R package version 3.1.0 (*Wickham, 2009*). Heatmaps were generated using pheatmap R package 1.0.10 (*Kolde, 2018*). The analysis scripts are available at: https://github.com/NordNeurogenomicsLab/Publications/tree/master/Canales_eLife_2021; *Canales, 2021*; copy archived at swh:1:rev:28836c8758908e130f1f6d8bfb3b6a112c6cbd1b.

## Differential expression analysis

Raw count data for all samples were used as input along with sample information for differential expression analysis using edgeR (*Robinson et al., 2010*). Genes with minimum $\log_2$ reads per kilo-base per million (RPKM) expression of −2 in at least two samples were included for analysis, result-ing in a final set of 17,195 genes for differential testing. For time-point differential expression analysis genes with count per million (CPM) >0.1 in at least two samples were considered, resulting in E12.5, 16,396; E14.5, 16,658; E17.5, 16,229; P0, 16,613 genes in differential expression analyses. Principal component analysis indicated that the strongest driver of variance across samples was developmental age. Tagwise dispersion estimates were generated, and differential expression analy-sis was performed with edgeR using a generalized linear model, separately for each developmental time-point, including sex, sequencing run factor, and treatment as the variable for testing. Normal-ized expression levels were generated using the edgeR rpkm function. Normalized $\log_2$ RPKM values were used for plotting of summary heatmaps and of expression data for individual genes.

## SFARI gene set enrichment analysis

The autism risk gene-set was downloaded from https://gene.sfari.org/ on 02-26-2019 and is included in this manuscript as *Supplementary file 17*. High confidence risk genes, annotated as 'gene-score' 1 and 2 were selected, and their orthologs were found using getLDS function from the biomaRt R package (*Durinck et al., 2009*; *Durinck et al., 2005*). Overlap of up- and downregulated DE genes with SFARI ortholog genes was calculated and plotted using a custom R script. Statistical signifi-cance of overlap was tested with hypergeometric test using the following R script: sum(dhyper(t:b, a, n - a, b)):

$t$ = number of overlapping DE genes with SFARI gene orthologs with gene_score 1 or 2
$b$ = number of SFARI gene orthologs with gene_score 1 or 2
$a$ = number of DE genes at a developmental stage
$n$ = number of genes at a developmental stage *sum* returns the sum of all the values present in its arguments. *dhyper* calculates density for the hypergeometric distribution.

## WGCNA

We used the WGCNA R package, version 1.66 (*Langfelder and Horvath, 2012*; *Langfelder and Horvath, 2008*) to construct signed co-expression networks using the entire dataset containing 24,015 genes. After the network construction, the gene set was filtered for minimal gene expression at an RPKM value of 0.25 or higher in at least two sample, resulting in a dataset consisting 17,195 genes. A correlation matrix using the biweight midcorrelation between all genes was computed for all relevant samples. The soft thresholding power was estimated and used to derive an adjacency matrix exhibiting approximate scale-free topology ($R^2$ >0.85). The adjacency matrix was transformed

to a topological overlap matrix (TOM). The matrix 1-TOM was used as the input to calculate co-expression modules using hierarchical clustering. Modules were branches of the hierarchical cluster tree base, with minimum module size set to 10 genes. Pearson's correlation coefficients were used to calculate correlation between sample traits (e.g., sex, treatment) and modules. The expression profile of a given module was summarized by the module eigengene (ME). Modules with highly correlated MEs (correlation >0.80) were merged together. The module connectivity (kME) of each gene was calculated by correlating the gene expression profile with module eigengenes. The module connectivity (kME) of each gene was calculated by correlating the gene expression profile with module eigengenes. Genes with no network correlation were placed into the module Grey. Following manual data inspection further highly correlated modules were merged: BrRePi: Brown, Red, and Pink, YeMaBl: Yellow, Magenta, and Black.

## Gene Ontology enrichment analysis

Mouse Gene Ontology (GO) data was downloaded from Bioconductor (org.Mm.eg.db). We used the TopGO R package version 2.34.0 (*Alexa and Rahnenfuhrer, 2019*) to test for enrichment of GO terms. For the analysis presented here, we restricted our testing to GO Biological Process annotations and required a minimal node size (number of genes annotated to GO terms) of 20. We used the internal 'weight01' testing framework and the Fisher test, a strategy recommended for gene set analysis that generally accounts for multiple testing comparisons. For GO BP analysis, we reported terms with p-value<0.05. For all enrichment analysis, the test set of DE genes was compared against the background set of genes expressed in our study based on minimum read-count cutoffs described above. Heatmaps showing positive $\log_2$ (expected/observed) values were plotted for GO terms of interest.

## Protein–protein interaction

Protein–protein interaction enrichment and network generation for E12.5 DE (FDR < 0.05) gene sets was performed using STRING (*Szklarczyk et al., 2019*), version 11, considering only experimentally and text mining interactions, with at least medium interaction confidence score. Disconnected nodes were removed from the network.

## Data analysis

For all experiments, data was collected from at least two (usually three, and sometimes more, as indicated in each analysis) experiments and is presented as the mean ± SEM. Protein validation data were analyzed by Student's *t*-test or one-way ANOVA, followed where appropriate by Tukey's honestly significant difference *post hoc* test (Graphpad Prism v.7). Significance was defined as: *p<0.05, **p<0.01, ***p<0.001, ****p<0.0001. Samples for RNA-seq were randomly collected across litters and processed blind to experimental condition. For differential gene expression analysis using edgeR, differences were considered statistically significant at high stringency for FDR < 0.05, or with reduced stringency for p values < 0.05.

## Acknowledgements

This work was supported by NIH NIMH (project # 5R21MH116681-02), NIH NIGMS (R35 GM119831), NARSAD Young Investigator Grant from Brain and Behavior Research Foundation and The UCD Clinical Translational Science Center. CPC is recipient of NIH NIMH Institutional National Research Service Award Autism Research Training Program fellowship (project # 2T32MH073124-16).

## Additional information

### Funding

| Funder | Grant reference number | Author |
|---|---|---|
| National Institute of Mental Health | 2T32MH073124-16 | Cesar P Canales |

| National Institute of Mental Health | 5R21MH116681-02 | Kim McAllister<br>Alex S Nord |
|---|---|---|
| Brain Research Foundation | | Alex S Nord |
| UCD | | Kim McAllister<br>Alex S Nord |
| National Institute of General Medical Sciences | GM119831 | Alex S Nord |
| National Institute of Mental Health | P50 MH106438 | A Kimberley McAllister |
| National Institute of Mental Health | T32MH112507 | Kathryn Prendergast |

The funders had no role in study design, data collection and interpretation, or the decision to submit the work for publication.

## Author contributions

Cesar P Canales, Conceptualization, Formal analysis, Validation, Visualization, Methodology, Writing - original draft, Project administration, Writing - review and editing; Myka L Estes, Conceptualization, Visualization, Methodology, Writing - original draft, Writing - review and editing; Karol Cichewicz, Conceptualization, Data curation, Software, Formal analysis, Visualization, Writing - original draft, Writing - review and editing; Kartik Angara, Linda Su-Feher, Formal analysis, Methodology; John Paul Aboubechara, Scott Cameron, Kathryn Prendergast, Iva Zdilar, Ellie J Kreun, Emma C Connolly, Jin Myeong Seo, Kathleen Farrelly, Tyler W Stradleigh, Deborah van der List, Lori Haapanen, Methodology; Jack B Goon, Data curation, Methodology; Judy Van de Water, Conceptualization, Resources, Supervision, Investigation, Methodology, Writing - review and editing; Daniel Vogt, Formal analysis, Supervision, Investigation, Writing - review and editing; A Kimberley McAllister, Conceptualization, Resources, Supervision, Funding acquisition, Investigation, Visualization, Project administration, Writing - review and editing; Alex S Nord, Conceptualization, Resources, Data curation, Software, Formal analysis, Supervision, Funding acquisition, Investigation, Visualization, Methodology, Writing - original draft, Project administration, Writing - review and editing

## Author ORCIDs

Cesar P Canales ⬤ https://orcid.org/0000-0003-2505-8367
Karol Cichewicz ⬤ https://orcid.org/0000-0001-5926-3663
A Kimberley McAllister ⬤ https://orcid.org/0000-0001-9177-9889
Alex S Nord ⬤ https://orcid.org/0000-0003-4259-7514

## Ethics

Animal experimentation: This study was conducted in compliance with NIH guidelines and approved protocols from the University of California, Davis Animal Care and Use Committee (IACUC) protocol #20229.

## Decision letter and Author response

Decision letter https://doi.org/10.7554/eLife.60100.sa1
Author response https://doi.org/10.7554/eLife.60100.sa2

# Additional files

## Supplementary files

• Supplementary file 1. Differential gene expression data of control vs MIA samples, and WGCNA gene module assignment at E12.5. Columns indicate gene symbols (gene_name), $\log_2$ fold change (LogFC), $\log_2$ counts per million (LogCPM), likelihood ratio statistics (LR), p value (PValue), false discovery rate adjusted p value (FDR), and WGCNA module assignment (moduleColors and moduleColors_merged). A total of 16,396 genes were tested.

• Supplementary file 2. Differential gene expression data of control vs MIA samples, and WGCNA gene module assignment at E14.5. Columns indicate gene symbols (gene_name), log$_2$ fold change (LogFC), log$_2$ counts per million (LogCPM), likelihood ratio statistics (LR), p value (PValue), false discovery rate adjusted p value (FDR), and WGCNA module assignment (moduleColors and moduleColors_merged). A total of 16,658 genes were tested.

• Supplementary file 3. Differential gene expression data of control vs MIA samples, and WGCNA gene module assignment at E17.5. Columns indicate gene symbols (gene_name), log$_2$ fold change (LogFC), log$_2$ counts per million (LogCPM), likelihood ratio statistics (LR), p value (PValue), false discovery rate adjusted p value (FDR), and WGCNA module assignment (moduleColors and moduleColors_merged). A total of 16,229 genes were tested.

• Supplementary file 4. Differential gene expression data of control vs MIA samples, and WGCNA gene module assignment at E12.5. Columns indicate gene symbols (gene_name), log$_2$ fold change (LogFC), log$_2$ counts per million (LogCPM), likelihood ratio statistics (LR), p value (PValue), false discovery rate adjusted p value (FDR), and WGCNA module assignment (moduleColors and moduleColors_merged). A total of 16,613 genes were tested.

• Supplementary file 5. RNA-seq sample metadata including Sample ID, age, treatment (Condition), classification as per their IL-6 prenatal levels (Response), sequencer lane batch parameters (HiSeq Lane, Library barcodes), animal and litter parameters (Animal ID, Litter ID, Liter Size), animal sex inferred from *Xist* expression (Sex by *Xist* counts), fastq file name, and fastq file md5sum hash value (md5sum). Out of 74 samples, 73 were tracked for the Litter ID (explained in the Comments).

• Supplementary file 6. Differential gene expression data of control vs MIA male samples at E12.5. Columns indicate gene symbols (gene_name), log$_2$ fold change (LogFC), log$_2$ counts per million (LogCPM), likelihood ratio statistics (LR), p value (PValue), and false discovery rate adjusted p value (FDR). A total of 16,171 genes were tested.

• Supplementary file 7. Differential gene expression data of control vs MIA male samples at E14.5. Columns indicate gene symbols (gene_name), log$_2$ fold change (LogFC), log$_2$ counts per million (LogCPM), likelihood ratio statistics (LR), p value (PValue), and false discovery rate adjusted p value (FDR). A total of 16,568 genes were tested.

• Supplementary file 8. Differential gene expression data of control vs MIA male samples at E17.5. Columns indicate gene symbols (gene_name), log$_2$ fold change (LogFC), log$_2$ counts per million (LogCPM), likelihood ratio statistics (LR), p value (PValue), and false discovery rate adjusted p value (FDR). A total of 15,883 genes were tested.

• Supplementary file 9. Differential gene expression data of control vs MIA male samples at P0. Columns indicate gene symbols (gene_name), log$_2$ fold change (LogFC), log$_2$ counts per million (LogCPM), likelihood ratio statistics (LR), p value (PValue), and false discovery rate adjusted p value (FDR). A total of 16,394 genes were tested.

• Supplementary file 10. Differential gene expression data of control vs MIA female samples at E12.5. Columns indicate gene symbols (gene_name), log$_2$ fold change (LogFC), log$_2$ counts per million (LogCPM), likelihood ratio statistics (LR), p value (PValue), and false discovery rate adjusted p value (FDR). A total of 16,161 genes were tested.

• Supplementary file 11. Differential gene expression data of control vs MIA female samples at E14.5. Columns indicate gene symbols (gene_name), log$_2$ fold change (LogFC), log$_2$ counts per million (LogCPM), likelihood ratio statistics (LR), p value (PValue), and false discovery rate adjusted p value (FDR). A total of 16,350 genes were tested.

• Supplementary file 12. Differential gene expression data of control vs MIA female samples at E17.5. Columns indicate gene symbols (gene_name), log$_2$ fold change (LogFC), log$_2$ counts per million (LogCPM), likelihood ratio statistics (LR), p value (PValue), and false discovery rate adjusted p value (FDR). A total of 16,093 genes were tested.

• Supplementary file 13. Differential gene expression data of control vs MIA female samples at P0. Columns indicate gene symbols (gene_name), log$_2$ fold change (LogFC), log$_2$ counts per million

(LogCPM), likelihood ratio statistics (LR), p value (PValue), and false discovery rate adjusted p value (FDR). A total of 16,487 genes were tested.

• Supplementary file 14. Differential gene expression data for all four developmental ages (Age) analyzed without sex covariate. Columns indicate gene symbols (gene_name), $\log_2$ fold change (LogFC), $\log_2$ counts per million (LogCPM), likelihood ratio statistics (LR), p value (PValue), and false discovery rate adjusted p value (FDR).

• Supplementary file 15. Differential gene expression data contrasting male vs female samples at each developmental timepoint (Age). Columns indicate gene symbols (gene_name), $\log_2$ fold change (LogFC), $\log_2$ counts per million (LogCPM), likelihood ratio statistics (LR), p value (PValue), and false discovery rate adjusted p value (FDR).

• Supplementary file 16. High confidence SFARI mouse orthologs genes used in the analysis. Columns indicate gene symbols and names, chromosomal location, genetic category, gene and syndromic scores, number of reports and Mouse Genome Informatics (MGI) symbols for mouse orthologs. 89 mouse orthologs with high confidence SFARI gene scores of one or two were included in the analysis.

• Supplementary file 17. Complete SFARI gene database available at the time of conducting the analysis.

• Supplementary file 18. Gene ontology biological process enrichment analysis for genes passing FDR < 0.05, for each developmental timepoint and direction of differential expression, summarized in *Figure 2d* heatmap. Columns signify gene ontology ID number (GO.ID), gene ontology term (Term), the number of annotated genes in the term (Annotated), the number of genes passing FDR < 0.05 for up- or downregulated genes (Significant), the expected number of genes under the null hypothesis, Fisher's p value (classicFisher), developmental age (Age), direction of differential expression relative to saline (DE_direction), enrichment.

• Supplementary file 19. Gene ontology biological process enrichment analysis for genes passing FDR < 0.05, for each developmental timepoint, direction of differential expression, and WGCNA module membership. Columns signify gene ontology ID number (GO.ID), gene ontology term (Term), the number of annotated genes in the term (Annotated), the number of genes passing FDR < 0.05 for up- or downregulated genes and WGCNA module (Significant), the expected number of genes under the null hypothesis, Fisher's p value (classicFisher), WGCNA module membership (module_color), enrichment, direction of differential expression (Direction), and developmental age (Age).

• Supplementary file 20. Numbers of RNA-seq samples divided by sex and treatment condition.

• Supplementary file 21. Summary of sample numbers (N) used for neuroanatomical IHC validation at E17.5. Numbers represent samples analyzed for at least one of the individual listed markers or co-stains. Details of specific use of individual samples can be found in *supplementary file 22*.

• Supplementary file 22. List of all MIA and control samples analyzed through neuroanatomical IHC validation at E17.5, and its specific use for individual experiments.

• Transparent reporting form

### Data availability

Raw and gene-count RNA-seq sequencing data are deposited in GEO under accession number GSE166376. Data analysis code is available at: https://github.com/NordNeurogenomicsLab/Publications/tree/master/Canales_eLife_2021, copy archived at https://archive.softwareheritage.org/swh:1:rev:28836c8758908e130f1f6d8bfb3b6a112c6cbd1b/ Detailed results of differential gene expression analysis are included in the manuscript and supporting files (Supplementary files 1-15), and can be visualized using our interactive online browser at: https://nordlab.shinyapps.io/MIA_RPKM_plots/.

The following datasets were generated:

| Author(s) | Year | Dataset title | Dataset URL | Database and Identifier |
|---|---|---|---|---|
| Cichewicz K, Nord | 2021 | A temporal map of maternal | https://www.ncbi.nlm. | NCBI Gene Expression |

| AS | immune activation-induced changes reveals a shift in neurodevelopmental timing and perturbed cortical development in mice | | nih.gov/geo/query/acc. cgi?acc=GSE166376 | Omnibus, GSE166376 |
| Cichewicz K, Nord AS | A temporal map of maternal immune activation-induced changes reveals a shift in neurodevelopmental timing and perturbed cortical development in mice | 2021 | https://github.com/NordNeurogenomicsLab/Publications/tree/master/Canales_eLife_2021 | NordNeurogenomicsLab, eLife_2021 |

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
