## [Decision Letter]

**Acceptance summary:**

The manuscript by Canales provides a comprehensive study of the effects of maternal immune activation (MIA) using a rigorous and carefully validated MIA mouse model. A major strength of the manuscript is the investigation of changes across time. The results are of broad interest to the fields of developmental neuroscience and neurodevelopmental disorders, where MIA is a risk factor for schizophrenia and autism.

**Decision letter after peer review:**

Thank you for submitting your article "Sequential perturbations to mouse corticogenesis following in utero maternal immune activation" for consideration by *eLife*. Your article has been reviewed by three peer reviewers, including Anita Bhattacharyya as the Reviewing Editor and Reviewer #1, and the evaluation has been overseen by Kathryn Cheah as the Senior Editor. The following individual involved in review of your submission has agreed to reveal their identity: Hanna Stevens (Reviewer #2).

The reviewers have discussed the reviews with one another and the Reviewing Editor has drafted this decision to help you prepare a revised submission.

Your manuscript provides a comprehensive study of the effects of maternal immune activation (MIA) using a rigorous and carefully validated MIA model. A major strength of the manuscript is the investigation of changes across time. The results are of broad interest to the fields of developmental neuroscience and neurodevelopmental disorders, where MIA is a risk factor for schizophrenia and autism. Despite these strengths, there are significant problems that limit the potential impact of the results. There are areas in which additional support is needed for the conclusions and the authors should be careful to state conclusions that are supported by specific experimental results.

Essential revisions:

(1a) The data in Figure 4 are compelling but do not yet definitively support a conclusion of reduced proliferation at E17.5 following MIA.

(1b) In addition, the authors should avoid hyperbole when describing these results ("MIA profoundly alters cortical proliferation.")

(1c) Additional quantification of cell types would bolster the results: The authors should quantify Ki67+ cells in the same manner that they quantified PH3+ cells and analyze expression of the progenitor marker SOX2, which tends to be more restricted to the VZ, in addition to PAX6 and SOX9.

(1d) In addition to measuring cortical thickness and counting SOX9+ and PH3+ cells, birth-dating can help support the under-proliferation/precocious neurogenesis hypothesis to test whether neural progenitor cells are exiting the cell cycle and differentiating early, migrating out of the VZ early, or both. Alternatively, an analysis of cell cycle kinetics would be useful; delayed mitosis and extended cell cycle time could contribute to the reduction in early-born cell types.

(1e) In the Discussion, the authors describe transcriptional signatures as "robust". What is the basis of this claim?

(2a) Additional analyses are needed to support the results of layer specific differences induced by MIA in in Figure 5.

(2b) Tbr1 thickness is not significantly different, as suggested in the text.

(2c) The authors should clarify whether Satb2 thickness measured using all Satb2+ layers or just upper layers.

(2d) A subset of layer 6 neurons expresses Ctip2, so it is hard to determine whether the phenotype represents a reduction in layer 6 or layer 5 Ctip2+ neurons, or a reduction in Ctip2+ neurons in general. The authors should co-label with Tbr1 and additional deep layer markers, such as Satb2 (which is expressed in a subset of layer 5 neurons), Fezf2, Sox5, and/or Rbp4 to bet a better idea of layer 6-layer 5 separation. The results do not rule out possibility that migration may be affected by MIA and the authors should address this point.

(3a) The Olig2 result suggests that later born cells are over-represented in the MIA cortex at the expense of younger-born cells, which could have a secondary effect on interneuron development. The authors should be careful to interpret this as a difference in progenitor maturation, not neuronal maturation.

(3b) In addition, other later-born cell type markers, such as Olig1 or Pdgfra, should be analyzed.

(4) Transcriptional analysis

(4a) Since the majority of variation is coming from developmental age, more subtle expression changes due to MIA might be missed by PCA and WGCNA.

(4b) The authors should discuss the definition of the "grey module" more deeply and discuss the fact that it is the only one that correlates with MIA.

(4c) The results that no DE genes with an FDR < 0.05 at P0 is surprising. The authors should address explanations for why expression appears to normalize suddenly, both biological explanations and technical (e.g., result of noisy or stochastic genes being falsely identified as DEGs).

(5a) A more nuanced acknowledgment of the different sources of tissue, in particular when it may have a bearing on the interpretations, is important.

(5b) The focus on cortical development is warranted and borne out in analysis at later timepoints but early timepoint tissue includes more than cortex.

(5c) RNA sequencing analysis could include tissue type as a covariate (total forebrain for E12.5 and dorsal forebrain for E14.5, E17.5, and P0).

(5d) Given that protein was taken from total forebrain at E17.5 for western analysis, the legends of the figures that show these results should not call this "cortical extracts" and the high expression of ctip2 in ventral forebrain at E17.5 should inform the reporting/interpretation of the results.

(5e) The differential development of the hippocampus should be considered. The authors should address whether the primordial hippocampus may be the prominent source of RNA difference and whether pax6 or Sox9 is elevated in that region, determining whether radial glia or newly differentiated astrocytes may be a source in this particular domain. This may reveal an interesting component of the developmental advancement, given the slightly different timing of hippocampal development relative to neocortex.

(6) The discussion of the enrichment of DE ASD related genes and the relationship to MIA is underdeveloped. What specific conclusions/parallels can be made?

(7) Additional details are needed to address rigor of analyses:

(7a) The authors should state explicitly how many biological replicates were used per condition per stage.

(7b) The authors should add a clustering analysis in Figure 1 (or move the PCA to Figure 1) to help make it more explicit.

(7c) Perhaps a uMAP or tSNE rather than simple PCA?

(7d) For immunostaining the authors should state whether the samples taken from the same litters as those samples used for RNA sequencing.

(7e) Parameters for cellular data and image analysis are not defined. No details are provided about number of sections and cells that were counted with ImageJ and how chosen fields were randomized.

---

## [Author Response]

Essential revisions:(1a) The data in Figure 4 are compelling but do not yet definitively support a conclusion of reduced proliferation at E17.5 following MIA.

We thank the reviewers for suggesting experiments to further elucidate proliferative phenotypes. As described below, we performed new IHC experiments for SOX2 and Ki67/PH3 labeling from the same animals and re-analyzed our initial results as recommended. The revised version of the manuscript better characterizes progenitor cells and provides stronger evidence of reduced mitosis at E17.5 following MIA (See response 1c below).

(1b) In addition, the authors should avoid hyperbole when describing these results ("MIA profoundly alters cortical proliferation.")

We removed the word “profoundly”. The title of the figure now reads:

“Figure 4. Altered cortical proliferation and lamination dynamics in fetal offspring five days after induction of MIA.”

(1c) Additional quantification of cell types would bolster the results: The authors should quantify Ki67+ cells in the same manner that they quantified PH3+ cells and analyze expression of the progenitor marker SOX2, which tends to be more restricted to the VZ, in addition to PAX6 and SOX9.

Following the review request, we quantified Ki67+ cells and performed new SOX2 IHC. The revised results are consistent with reduced progenitor cell findings we reported in the initial submission, with the new data improving the strength and rigor of this analysis. Specifically, we found significantly reduced Ki67+ cell counts and reduced VZ-specific Sox2^+^ cell counts in in MIA-exposed group. The new data has been incorporated into Figure 4 and is described in the revised text below:

“To characterize the neuroanatomical specificity of Blue and Turquoise dysregulation transcriptional signatures (Figure 4D), we performed a static immuno-fluorescent labeling comparison of late neurogenesis, cortical lamination and cell specification at E17.5 in an independent MIA cohort. E17.5 represents the end of the peak of neurogenesis during cortical development, before the neuronal-glial transition that occurs around E18.5. […] Reduced expression of SOX2, a progenitor marker that is restricted to the VZ, was also observed in MIA brains (SOX2 VZ, P=0.0006 two tailed Student’s t-test) (Figure 4F)”.

“To determine if the decrease in progenitor markers might coincide with decreased proliferation, we next labeled cells undergoing proliferation, using the marker KI67 (any stage of cell proliferation) and the Phospho Histone 3 (PH3) marker, which labels a subset of cycling cells in mitosis (M) phase (Hendzel et al., 1997; Walton et al., 2019). […] Moreover, within the reduced progenitor cell populations at E17.5, the number of cells in the mitosis stage of the cell cycle are decreased to a greater extent.”

1d) In addition to measuring cortical thickness and counting SOX9+ and PH3+ cells, birth-dating can help support the under-proliferation/precocious neurogenesis hypothesis to test whether neural progenitor cells are exiting the cell cycle and differentiating early, migrating out of the VZ early, or both. Alternatively, an analysis of cell cycle kinetics would be useful; delayed mitosis and extended cell cycle time could contribute to the reduction in early-born cell types.

We agree that experiments such as birth dating, analysis of cell cycle timing and exit, and analysis of other time-points across the window of neurogenesis will be needed to ultimately resolve the impacts of MIA across neurogenesis and gliogenesis. Regrettably, such in depth analysis is beyond the scope of this manuscript. Nonetheless, we were able to follow the alternative strategy proposed in the review summary, providing an initial perspective on cell cycle kinetics via comparing the ratio of overall cycling cell numbers (Ki67+) with cells undergoing mitosis and labeled by Phospho Histone 3 (PH3), which is present in M phase. In addition to an overall decrease in VZ progenitor cell populations, the reduced Ki67/PH3 ratio revealed a significantly decreased proportion of progenitor cells undergoing mitosis in MIA brains at E17.5. This demonstrates that cell cycle kinetics and mitosis are altered beyond a simple reduction in progenitor cells at E17.5. The altered ratio is consistent with reduced time in M phase and/or a decreased number of cycling progenitors. This result is exciting and extends our findings. In addition to describing the new finding, we additionally acknowledge in the Discussion section that birth-dating, cell cycle kinetics, and time course experiments are needed to comprehensively determine altered proliferative and neurogenesis dynamics following MIA. The new experiments and analysis are included in Figure 4. The following text has been modified in the revised manuscript. To avoid reproduction, see text in [1c] as well as below.

“Although these findings are indicative of overall decreased neurogenesis by E17.5, specific birth-dating and time course experiments are needed to test whether progenitor cells exit the cell cycle differently and if overall neurogenesis is perturbed beyond the E17.5 time-point reported here.”

(1e) In the Discussion, the authors describe transcriptional signatures as "robust". What is the basis of this claim?

We describe the transcriptional signatures as “robust” because at E14.5 and E17.5 a large number of genes in our differential expression analysis surpass stringent statistical significance of FDR < 0.05, and substantial fold change (FC) difference, with many genes exceeding 2 FC (log2 = 1) or even 4 FC (log2 = 2) (Figure 1B-D). We acknowledge that the term robust is non-specific, but thought it captured the overall results. We are happy to substitute for another word or drop if the reviewers prefer, but we left this phrasing for now.

(2a) Additional analyses are needed to support the results of layer specific differences induced by MIA in in Figure 5.

We thank the reviewers for the suggested revisions to better define lamination phenotypes. In addition to clarifying the methods and more accurately describing the results, we performed new IHC experiments with TBR1 and SATB2 co-staining, as requested, and re-analyzed cell count and layer thickness data for consistency. The new IHC results further define differences in in MIA-exposed fetal cortex and provide evidence that the altered lamination is associated with differences in neuronal migration.

(2b) Tbr1 thickness is not significantly different, as suggested in the text.

The thickness of the Tbr1+ layer is not significantly decreased but the number of Tbr1+ cells is different. We apologize for this error. We made the following changes to the text:

“The number of TBR1+ and CTIP2+ cells, but not SATB2+ cells, were also significantly reduced in the neocortex from MIA offspring (TBR1+ cells, P = 0.0138; CTIP2+ cells, P = 0.0119; SATB2+ cells, P = 0.306, two tailed Student t-test) (Figure 5E)”

And:

“We found a significant reduction in CTIP2 layer thickness and a similar trend in TBR1 layer thickness in the MIA brains.”

(2c) The authors should clarify whether Satb2 thickness measured using all Satb2+ layers or just upper layers.

For consistency, all thickness analyses (including SATB2) were performed by measuring all positive cells for each marker. We clarify how these layer specific thickness analyses were performed when introducing these assays in the Results and added details to the Materials and methods section under “Gross anatomy, immunohistochemistry and image quantification analysis”:

“Examination of the distribution of the lamination markers TBR1 (layer 6), CTIP2 (layer 5 and a subset of cells in layer 6), SATB2 (layers 2-4, and a subset of cells in layer 5), and CUX1 (layers 2 and 3), revealed MIA-induced cortical layer-specific alterations in lamination, thickness, and cell density at E17.5 (Figure 5A-B, E-F).”

“Laminar thickness analyses were performed by measuring the spread of all positive cells within the cortical plate per individual marker.”

In Figure 5 legend: “(F) laminar thickness analysis, based on the spread of all labeled cells, are shown.”

(2d) A subset of layer 6 neurons expresses Ctip2, so it is hard to determine whether the phenotype represents a reduction in layer 6 or layer 5 Ctip2+ neurons, or a reduction in Ctip2+ neurons in general. The authors should co-label with Tbr1 and additional deep layer markers, such as Satb2 (which is expressed in a subset of layer 5 neurons), Fezf2, Sox5, and/or Rbp4 to bet a better idea of layer 6-layer 5 separation. The results do not rule out possibility that migration may be affected by MIA and the authors should address this point.

We are grateful for the reviewer’s point that Ctip2 is not a definitive marker distinguishing layer 5 or 6, and we have clarified this in the Figure 5 legend. As requested, we performed new experiments co-staining for TBR1 and SATB2 to further investigate the layer 5/6 lamination phenotype via labeling SATB2+ putative upper layer cells that form layers 2-4, as well as some Layer 5 cells. These experiments show that SATB2+ cells exhibit similar differences in positioning and differential overlap with Tbr1+ cells as Ctip2+ cells. These findings indicate generalized perturbation to lamination, rather than a specific impact on Ctip2+ layer 5/6 cells. Our new experiments show that SATB2+ cells in the MIA group are already in their more superficial position and show a more rounded morphology, indicating a more complete stage of lamination and further progression of migration of SATB2+ cells. These findings provide further evidence of lamination defects, and are consistent with our general interpretation that MIA alters developing cortical anatomy.

Of interest and relevant to the reviewer comment on migration, the new co-staining experiments also provide evidence of alterations in migration. By analyzing higher magnification images, we identified differences in cell body shape of both TBR1 and SATB2 positive cells between the MIA and control groups, indicative of an active radial migration state observable in control animals versus a more static rounded cell body shape in the MIA brains (shown in Figure 5D). This finding is consistent with our hypothesis that timing of lamination is altered following MIA exposure, with greater number of cells that were still migrating evident in the control versus MIA group.

These new TBR1+/SATB2+ co-labeling experiments are presented in Figure 5C-D and described in the Results and Discussion sections as follows:

“Examination of the distribution of the lamination markers TBR1 (layer 6), CTIP2 (layer 5 and a subset of cells in layer 6), SATB2 (layers 2-4, and a subset of cells in layer 5), and CUX1 (layers 2 and 3), revealed MIA-induced cortical layer-specific alterations in lamination, thickness, and cell density at E17.5 (Figure 5A-B, E-F). […] In contrast to previous reports (Choi et al., 2016), we did not find obvious cortical dysplasia or gross alterations in brain anatomy in the analyzed rostral and caudal regions, including somatosensory cortex and S1DZ (Figure 5E-F).”

(3a) The Olig2 result suggests that later born cells are over-represented in the MIA cortex at the expense of younger-born cells, which could have a secondary effect on interneuron development. The authors should be careful to interpret this as a difference in progenitor maturation, not neuronal maturation.

We agree that the cell-type specific impacts could be either primary or secondary, and are likely to be driven by complex interactions between cell-autonomous and non-cell autonomous signaling. Examining the impact of MIA on specific neuronal and glial populations (including interneurons) has been a focus of other published studies that we reference. We hope our findings spur further future work. We also agree that this should be referred to as progenitor maturation, and updated the text accordingly. The text below from the Discussion now addresses this point:

“At this developmental stage, our data suggest that MIA induces an acute expenditure of the neocortical, and potentially other, progenitor pools, which in turn leads to an expedited maturation of cells that would normally be generated later during development. […] Although these findings are indicative of overall decreased neurogenesis by E17.5, specific birth-dating and time course experiments are needed to test whether progenitor cells exit the cell cycle differently and if overall neurogenesis is perturbed beyond the E17.5 time-point reported here.”

(3b) In addition, other later-born cell type markers, such as Olig1 or Pdgfra, should be analyzed.

Following the reviewer’s recommendation, we added new IHC experiments staining for PDGRFa. We include quantification analysis by zones for this new marker, following the same strategy we used for DLX2 in the initial submission. PDGFRa is specific to immature oligodendrocytes and showed similar over-representation as seen in our original OLIG2 IHC analysis. These new experiments are presented as Figure 6E, F, Ll. The paragraph in the text that describes these new findings is below:

“To investigate other cell populations that may be impacted by MIA during development, we analyzed astrocytes, oligodendrocytes and GABAergic interneuron cell populations. […] Thus, these experiments provide evidence that MIA leads to increased numbers of additional cell types (putative astrocytes and oligodendrocytes) that are typically born at later ages during corticogenesis, reinforcing findings of accelerated cell development that is supported by an increased representation of later-born cell types at E17.5 in brains exposed to MIA.”

(4) Transcriptional analysis(4a) Since the majority of variation is coming from developmental age, more subtle expression changes due to MIA might be missed by PCA and WGCNA.

This concern seems to be based on some misunderstanding of our approach to identify and characterize MIA transcriptional signatures. We fully agree that MIA-associated differential expression signatures will not be adequately captured by PCA and WGCNA. While PCA and WGCNA are powerful approaches, they are not used as primary methods for differential expression analysis. Indeed, we use these tools as secondary in the context of contextualizing MIA-associated differential expression (DE) signatures at a systems level. For primary DE analysis and identification of DE genes, we used a well-established method that maximizes power via controlling for co-variates, the generalized linear model framework implemented in edgeR. We have added text to clarify this strategy:

“For transcriptomic analysis, we used a strategy of first defining DE genes at each time-point using the general linear model (GLM) approach implemented in edgeR, followed by mapping DE signatures to systems-level expression patterns via module assignment using weighted gene co-expression network analysis (WGCNA)”

As described in the manuscript, we applied a pairwise DE testing strategy, examining RNA-seq each time-point independently and including technical batch and sex as covariates in the GLM. We also tested a full model, including all samples with time-point as a covariate, but feel this model inflated significance values for some genes that were DE across the time course and missed stage-specific changes. We also performed DE analysis on sex-stratified samples, i.e. independently on males and females, as reported in the manuscript. This DE analysis, gene ontology enrichment analysis, and modeling of the developmental timing are reported in the Results and figures (Figure 1D-F, Figure 4C).

Our use of PCA was for initial insights into overall variance across all samples, as is common in RNA-seq studies. This dimensionality reduction approach is useful for detecting technical irregularities (quality control) and for visualizing major components of variation across samples. In this analysis, PC1 and PC2 captured variation associated with developmental age, with some further stratification between MIA and saline samples. We now include these PCA plots as Figure 1B, as requested. We later took advantage of the utility of PC1 and PC2 for capturing developmental age to build an age-predictive model to statistically test the observation that MIA samples at E14.5 and E17.5 exhibit differences in predicted age, as reported in the Results and presented in Figure 4. PCA itself was not used to determine or test differential expression.

Finally, we used weighted gene co-expression network analysis (WGCNA) as a method used for identifying module patterns and contextualizing and grouping DE genes at the systems/module level. We describe how DE genes map to WGCNA co-expression modules in Figure 2 and focus on DE genes and specific modules in Figures 3-5. This approach enabled association of DE genes to major acute and secondary transcriptional changes. Overall, our paired use of DE (here via GLM) and WGCNA is a standard approach in the field to link gene-level DE changes with systems-level signatures in transcriptomics data.

(4b) The authors should discuss the definition of the "grey module" more deeply and discuss the fact that it is the only one that correlates with MIA.

We first want to clarify that Grey and Green modules are both significantly associated with the MIA phenotype, and YeMaBl as well demonstrates marginal significance. Thus, the review comment that the Grey module was the only MIA-associated module is incorrect. This was described in the following section of the Results in the initial manuscript: “The WGCNA-resolved modules enabled mapping of DE signatures to neurodevelopmental processes that were acutely induced by MIA. There was significant association between genes in Grey (P = 0.007) and Green (P = 0.03) modules and MIA treatment, and marginal significance with MIA for the YeMaBl (P = 0.05) module (Figure 2—figure supplement 1B).” We go into depth interrogating these three MIA associated modules in the Results text, in Figure 3, and in the Discussion.

Nonetheless, we acknowledge the general point that it is interesting that the Grey module was associated with MIA at the trait level considering this module is made up of the unassigned genes that did not exhibit strong co-expression with the other modules. We explored this finding during our initial analysis to ensure that this was not due to WGCNA parameters. In other words, we tested if the MIA-associated genes in Grey would be assigned to a specific new module if we changed the module assignment parameters. When we increased the number of modules, the composition of Grey shifted, but the association with MIA remained. Further, increasing the number of modules resulted in over-splitting and some reduced biological coherence of individual modules. We thus elected to use WGCNA parameters that generated modules that effectively captured general trends of developmental expression as well as MIA-associated signatures, as reported in the manuscript. In response to the reviewer’s point, we added text to the Discussion regarding the Grey-MIA association:

“Significant association of the Grey unassigned module with the MIA phenotype was driven by a subset of genes that showed strong acute induction at E12.5 and E14.5. […] Thus, we opted for reduced splitting to capture major co-expression trends.”

(4c) The results that no DE genes with an FDR < 0.05 at P0 is surprising. The authors should address explanations for why expression appears to normalize suddenly, both biological explanations and technical (e.g., result of noisy or stochastic genes being falsely identified as DEGs).

We agree that it was somewhat surprising that the P0 DE signatures were weaker at the gene level compared to the other time-points. This result does not appear to be due to our experimental approach, since the P0 time-point was based on similar sample numbers, sex distribution, batch complexity similar, sequencing depth, and general quality metrics as for the other time-points in our study. In our original manuscript, we speculated that the reduced signature at P0 supports that the significant DE signatures at the earlier time-points reflect transient perturbation at the transcriptional level. As we found strong reproducible relationships between DE signatures at E12.5 and E14.5 and between E14.5 and E17.5, we believe that the MIA signatures of these time-points are unlikely to be false positives. As noted in the review, we agree with the possibility that some unknown technical issue or that general increased biological variability could both reduce the power to detect P0 DE in our experiments. Pursuing the time course approach beyond P0 in future work will be useful to understand the continuing effects, but such further investigation is beyond the scope of this manuscript. We updated the relevant sentence in the Discussion to incorporate suggestions made in the reviewer critique:

“While significantly subtler in overall effects compared to earlier time-points, we also see evidence for altered neuronal transcription and lasting immune activation at P0. […] Nonetheless, the subtle changes in gene expression at P0 are consistent with deficits in synapse formation that require changes in expression of immune molecules in neurons from newborn MIA offspring (Elmer et al., 2013).”

(5a) A more nuanced acknowledgment of the different sources of tissue, in particular when it may have a bearing on the interpretations, is important.

We acknowledge the importance of communicating our dissection strategy and source of tissue in order to clearly interpret RNA-seq results. We attempted to clarify the methods and add discussion to address the relevant points below.

(5b) The focus on cortical development is warranted and borne out in analysis at later timepoints but early timepoint tissue includes more than cortex.

First, we apologize that the description of our dissection methods was not specific enough and have clarified the Materials and methods text in the revised manuscript (see below). We have also made clear in the Discussion that some subpallial tissues may have been collected in addition to cortex at the youngest age examined, and that this difference in approach may drive some aspects of the differential expression signature (see text below). Finally, we clarify that our strategy at E12.5 was to collect mainly dorsal telencephalon/pallium, but that our approach likely also captured some ventral telencephalon/sub-pallial structures as well as superficial tissue.

“Tissue dissections for RNA-seq and Immunoblotting. Dissections of dorsal telencephalon/pallium or developing cortex were performed at E12.5 + 6 hrs, E14.5, E17.5, and P0 from control (saline) and poly(I:C)-exposed MIA litters. At E12.5, due to the smaller embryo size, dissections were made via a diagonal cut that maximized collection of dorsal telencephalon, but could have also included some subpallium/ventral telencephalon and superficial tissues. […] Independent litters were used for RNA-seq and immunoblotting.”

“Finally, bulk tissue RNA-seq is limited in cell type-specific resolution and signatures could additionally be in part driven by cortex-adjacent structures, such as primordial hippocampus, that may have also been captured at the youngest age. Future work using single cell approaches is needed to resolve the impacted cell types and compare transcriptional pathology following MIA.”

(5c) RNA sequencing analysis could include tissue type as a covariate (total forebrain for E12.5 and dorsal forebrain for E14.5, E17.5, and P0).

All differential expression analysis was performed independently for each developmental stage via comparison of matched MIA and control samples. Thus, all samples considered in each DE GLM analysis and the same tissue make up and same developmental stage, precluding the inclusion of any covariate. Within stage, dissections were performed blinded to condition and variation across samples should be random and thus averaged out across replicates. We hope this clarifies the method and addresses this comment. To clarify this in the manuscript, we describe these methods as follows:

“Tagwise dispersion estimates were generated, and differential expression analysis was performed with edgeR using a generalized linear model, separately for each developmental time-point, including sex, sequencing run factor, and treatment as the variable for testing.”

(5d) Given that protein was taken from total forebrain at E17.5 for western analysis, the legends of the figures that show these results should not call this "cortical extracts" and the high expression of ctip2 in ventral forebrain at E17.5 should inform the reporting/interpretation of the results.

We apologize for this error in the methods. The dissection strategy for RNA-seq and immunoblotting sample collection at E17.5 was identical and only includes developing cortex. Thus RNA and protein quantitative analyses were performed on tissues collected using the same methods. We reorganized the Materials and methods section to highlight this by changing the subsection to the subtitle “Dissection methods for RNA-seq and immunoblotting.” We also removed the term “cortical extracts” from the legend for Figure 4—figure supplement 1A. See response text in [5b] for specific wording.

(5e) The differential development of the hippocampus should be considered. The authors should address whether the primordial hippocampus may be the prominent source of RNA difference and whether pax6 or Sox9 is elevated in that region, determining whether radial glia or newly differentiated astrocytes may be a source in this particular domain. This may reveal an interesting component of the developmental advancement, given the slightly different timing of hippocampal development relative to neocortex.

These questions regarding hippocampal contributions are interesting. Considering the dissection strategy of targeting pallium (due to challenge of isolating cortex), the primordial hippocampus was included and differences in this structure could contribute to our differential expression findings. While there may be some impact from structures such as early hippocampus (or alternatively ventral telencephalon as raised earlier), we believe the general findings do strongly implicate the pallium/cortex as the major source of DE signatures and importantly, our IHC results verified phenotypes in the developing cortex. Future studies examining region-specific differences across non-cortical structures are certainly of interest, though we believe they are outside the scope of our study. Nonetheless, we recognize the limits of manual dissection and bulk tissue strategies, and have added a clarifying sentence into the Discussion – see [5b] response text.

(6) The discussion of the enrichment of DE ASD related genes and the relationship to MIA is underdeveloped. What specific conclusions/parallels can be made?

We added text to the Discussion to further discuss this relationship. We note that the overlap was general, included both up- and down-regulated DE genes, and that overlapping ASD risk genes were annotated across biological pathways rather than specific to any particular function or process. Thus, while we think this is indicative of general translational relevance, we don’t see a clear conclusion beyond the general overlap implicating shared processes across corticogenesis. Thus, we tried to offer some further thoughts, but want to avoid speculation too far beyond the evidence. If the reviewers had further suggestions for how to summarize, we are open to amending.

“Among MIA-associated DE up-regulated and down-regulated genes, there was strong enrichment for ASD-relevant loci, indicating the general translational relevance of our MIA model to ASD.”

“The majority of the ASD-relevant genes that were DE following MIA (Figure 1E) were associated with perturbation to the modules associated with neurodevelopmental processes. […] These overlapping genes represent targets of interest for future studies investigating parallel mechanisms between genetic and MIA-associated NDD risk.”

(7) Additional details are needed to address rigor of analyses:(7a) The authors should state explicitly how many biological replicates were used per condition per stage.

In the Results section we added the following references to the supplementary files, which provide details of the sample size split by sex (Supplementary file 5), as well as a complete RNA-seq metadata spreadsheet (Supplementary file 20). The following underlined sentence detailing our sample size was also added to the Results section:

“RNA-seq datasets were generated from male and female embryos from 28 independent litters across control and MIA groups, typically with 1-3 embryos represented per independent litter (Supplementary file 20, Supplementary file 5). […] For transcriptomic analysis, we used a strategy of first defining DE genes at each time-point using the general linear model (GLM) approach implemented in edgeR, followed by mapping DE signatures to systems-level expression patterns via module assignment using weighted gene co-expression network analysis (WGCNA).”

(7b) The authors should add a clustering analysis in Figure 1 (or move the PCA to Figure 1) to help make it more explicit.

PCA plots are now included in Figure 1.

(7c) Perhaps a uMAP or tSNE rather than simple PCA?

We respectfully note that we report results of a bulk RNA-seq experiment and use the PCA solely to capture major components of variation. uMAP and tSNE are more sophisticated dimensionality reduction techniques used primarily to deal with visualization of complex, and most commonly single cell, relationships. With our simpler data, we are unclear what the benefit would be of using these methods for further exploration of sample relationships. Perhaps incorporating PCA plots into Figure 1, as requested, should clarify how the PCA results were used and interpreted and address this point.

(7d) For immunostaining the authors should state whether the samples taken from the same litters as those samples used for RNA sequencing.(7e) Parameters for cellular data and image analysis are not defined. No details are provided about number of sections and cells that were counted with ImageJ and how chosen fields were randomized.

We thank the reviewers for bringing this to our attention, we now realize that although all numbers were included for all experiments in our original submission, the presentation of this information lacked consistency and could be confusing. The following modifications were made to address this concern in our revised manuscript:

First, we further clarified the samples used for IHC in two new supplementary files. We note that since we attempted to use similar tissue sections for consistency, there was a limit on the number of analyses that could be done on individual samples. Further, we used some new litters/embryos for the experiments in the revision. Thus, we provide a simple summary in Supplementary file 21 including sample numbers (N) used for neuroanatomical IHC validation at E17.5, where numbers represent samples analyzed for at least one of the individual listed markers or co-stains. Details of specific use of individual samples is reported in Supplementary file 22.

Second, we added missing information regarding the imaging quantification methods. We apologize for this omission. We have now added details of parameters used for cellular counting data to the Materials and methods section and clarified that the analysis was performed in triplicate sections. Please see below our modified text, which is now part of the revised Materials and methods subsection “Gross anatomy, immunohistochemistry and image quantification analyses”:

“Histology was performed in triplicate sections in brains using male and female pups from at least two litters. Sections were selected to focus on the developing somatosensory cortex. Details on sample individual sample usage for each IHC experiment is detailed in Supplementary files 21 and 22.”

“For cell counting, boxes were drawn on specific areas of the neocortex for progenitor and mitosis markers; SOX2, SOX9 and PH3. This box was 300 x 300 pixels on an image of the neocortex obtained using a 20x objective. To be consistent, the ventral aspect of the box was drawn along the VZ and one corner touched the medial aspect of the VZ. The same strategy was used to draw boxes from Ki67 images except the box was increased in size to 500 x 500 pixels to account for the additional Ki67+ cells seen in neocortical areas above the SVZ. SOX9 boxes were drawn along the dorsal border edge (600 pixels in width) of the neocortex and extended ventrally two thirds the length of the neocortex in order to avoid the VZ/SVZ region. For laminar zone counts, a box spanning 300 pixels was drawn along the VZ and the dorsal/ventral proportions were equally divided into 3 zones. These box sizes allowed for a sufficient number of cells to be counted from each tissue. Laminar thickness analyses were performed by measuring the spread of all positive cells within the cortical plate per individual marker. For all quantification analyses, 1-3 sections per brain utilizing the developing hippocampus as a landmark for positioning were used.”